

# ScholarLens: extracting competences from research publications for the automatic generation of semantic user profiles

Bahar Sateli[1], Felicitas Löffler[2], Birgitta König-Ries[2] and René Witte[1]

[1] Semantic Software Lab, Department of Computer Science and Software Engineering, Concordia University, Montreal, Quebec, Canada

[2] Heinz-Nixdorf-Chair for Distributed Information Systems, Department of Mathematics and Computer Science, Friedrich Schiller University Jena, Jena, Germany

## ABSTRACT

**Motivation**. Scientists increasingly rely on intelligent information systems to help them in their daily tasks, in particular for managing research objects, like publications or datasets. The relatively young research field of *Semantic Publishing* has been addressing the question how scientific applications can be improved through semantically rich representations of research objects, in order to facilitate their discovery and re-use. To complement the efforts in this area, we propose an automatic workflow to construct *semantic user profiles* of scholars, so that scholarly applications, like digital libraries or data repositories, can better understand their users' interests, tasks, and competences, by incorporating these user profiles in their design. To make the user profiles sharable across applications, we propose to build them based on standard semantic web technologies, in particular the Resource Description Framework (RDF) for representing user profiles and Linked Open Data (LOD) sources for representing competence topics. To avoid the *cold start* problem, we suggest to automatically populate these profiles by analyzing the publications (co-)authored by users, which we hypothesize reflect their research competences.

**Results**. We developed a novel approach, *ScholarLens*, which can automatically generate semantic user profiles for authors of scholarly literature. For modeling the competences of scholarly users and groups, we surveyed a number of existing linked open data vocabularies. In accordance with the LOD best practices, we propose an RDF Schema (RDFS) based model for competence records that reuses existing vocabularies where appropriate. To automate the creation of semantic user profiles, we developed a complete, automated workflow that can generate semantic user profiles by analyzing full-text research articles through various natural language processing (NLP) techniques. In our method, we start by processing a set of research articles for a given user. Competences are derived by text mining the articles, including syntactic, semantic, and LOD entity linking steps. We then populate a knowledge base in RDF format with user profiles containing the extracted competences. We implemented our approach as an open source library and evaluated our system through two user studies, resulting in mean average precision (MAP) of up to 95%. As part of the evaluation, we also analyze the impact of semantic zoning of research articles on the accuracy of the resulting profiles. Finally, we demonstrate how these semantic user profiles can be applied in a number of use cases, including article ranking for personalized search and finding scientists competent in a topic —e.g., to find reviewers for a paper.

Corresponding author
Bahar Sateli,
sateli@semanticsoftware.info

**Availability**. All software and datasets presented in this paper are available under open source licenses in the supplements and documented at http://www.semanticsoftware. info/semantic-user-profiling-peerj-2016-supplements. Additionally, development releases of ScholarLens are available on our GitHub page: https://github.com/ SemanticSoftwareLab/ScholarLens.

# INTRODUCTION

Researchers increasingly leverage intelligent information systems for managing their research objects, like datasets, publications, or projects. An ongoing challenge is the overload scientists face when trying to identify relevant information, for example when using a web-based search engine: while it is easy to find numerous *potentially* relevant results, evaluating each of these is still performed manually and thus very time-consuming.

We argue that smarter scholarly applications require not just a semantically rich representation of research objects, but also of their users: By understanding a scientist's interests, competences, projects and tasks, intelligent systems can deliver improved results, e.g., by filtering and ranking results through personalization algorithms (*Sieg, Mobasher & Burke, 2007*).

So-called *user profiles* (*Brusilovsky & Millán, 2007*) have been adopted in domains like e-learning, recommender systems or personalized news portals (we provide a brief background on user profiling in the 'Background'). Increasingly, they also receive more and more attention in scientific applications, such as expertise retrieval systems. Constructing such user models automatically is still a challenging task and even though various approaches have already been proposed, a semantic solution based on Linked Open Data (LOD) (*Heath & Bizer, 2011*) principles is still missing.

We show that a semantically rich representation of users is crucial for enabling a number of advanced use cases in scholarly applications. One of our central points is that a new generation of *semantic user profile* models are ideally built on standard semantic web technologies, as these make the profiles accessible in an open format to multiple applications that require deeper knowledge of a user's competences and interests. In the 'Literature Review', we analyze a number of existing LOD vocabularies for describing scholars' preferences and competences. However, they all fall short when it comes to modeling a user's varying degrees of competence in different research topics across different projects. We describe our solution for scholarly user models in the 'Design'.

Bootstrapping such a user profile is an infamous issue in recommendation approaches, known as the *cold start* problem, as asking users to manually create possibly hundreds of entries for their profile is not realistic in practice. Our goal is to be able to create an accurate profile of a scientist's *competences*, which we hypothesize can be automatically calculated

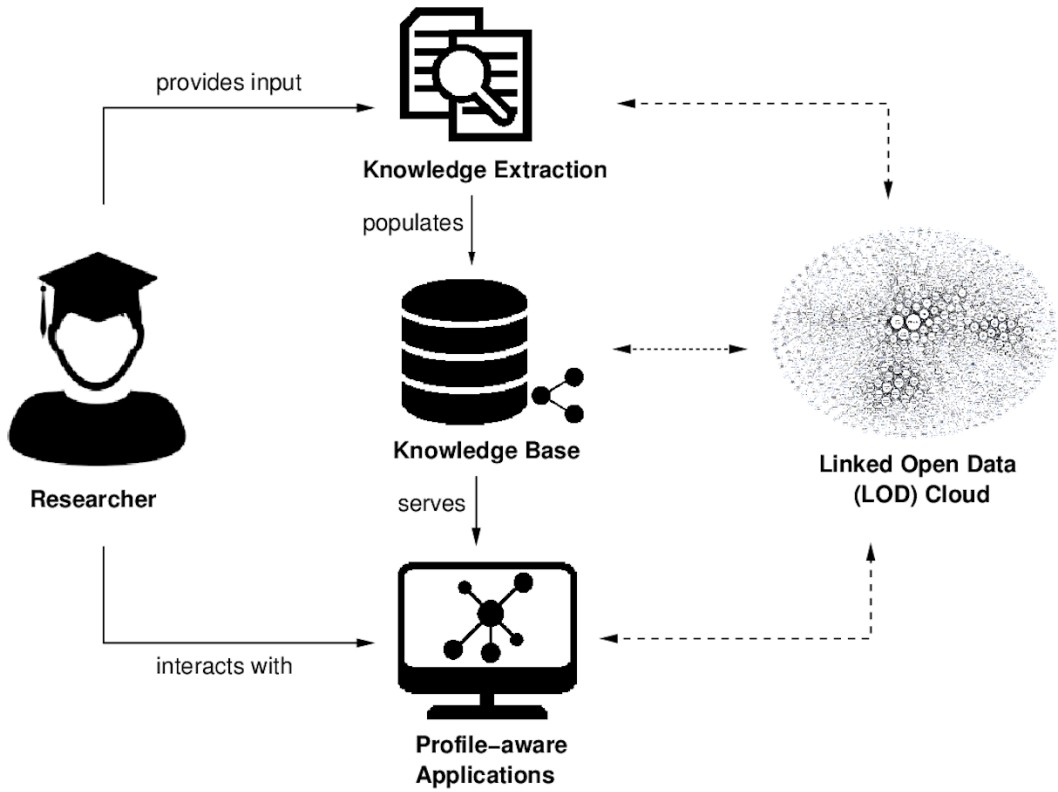

**Figure 1** This diagram shows a high-level overview of our approach to semantic user profiling: users can bootstrap their profiles by providing a set of their (co-)authored publications. The extracted knowledge is then stored in a knowledge base that can be incorporated in various scholarly applications. Researchers can then obtain personalized services through applications leveraging the semantic user profiles.

based on the publications of each user. Towards this end, we developed a novel NLP-based method for analyzing full-text research articles to extract an author's competences and constructing semantic user profiles in a linked open data format for automatic knowledge base construction. Our ideas are implemented in *ScholarLens* library, which we believe is the first open source library that facilitates the automatic construction of scholarly user profiles. The design and implementation of ScholarLens are detailed in 'Design' and 'Implementation', respectively. A high-level overview of our approach is illustrated in Fig. 1.

To evaluate our profile generation approach, we performed two user studies with ten and twenty-five scientists from various research groups across Europe and North America. The participants were provided with two different user profiles each, which were automatically generated based on their publications: one based on the articles' full texts, the second restricted to rhetorical entities (REs), like the *claims* and *contributions* in a paper (*Sateli & Witte, 2015*). In each study, we asked the participants to evaluate the generated top-N competence entries in their user profiles. The results, provided in the 'Evaluation', show that our approach can automatically generate user profiles with a precision of up to 95% (Mean Average Precision for top-10 competences).

Finally, we illustrate in the 'Application Examples' how semantic user profiles can be leveraged by scholarly information systems in a number of use cases, including a competence analysis for a user (e.g., for finding reviewers for a new paper) and re-ranking of article search results, based on a user's profile.

## BACKGROUND

In this section, we provide background information on user profiling, competence management and its applications. We also briefly introduce semantic publishing and its connections with natural language processing (NLP) techniques.

### User profiling and personalization

A user profile is an instance of a user model that contains either a user's characteristics, such as knowledge about a topic, interests and backgrounds, or focuses on the context of a user's work, e.g., location and time (*Brusilovsky & Millán, 2007*). Depending on the application offering personalized content, different features have to be taken into account when modeling user profiles. For instance, educational learning systems typically model a user's knowledge and background, whereas recommender systems and search applications are more focused on a user's interests. Constructing user profiles requires collecting user information over an extended period of time. This gathering process is called *user profiling* and distinguishes between *explicit* and *implicit* user feedback (*Gauch et al., 2007*). Explicit user feedback actively requests interests from a user, whereas implicit user feedback derives preferences from the user's activities. Commonly used implicit profiling techniques observe the user's browsing behavior and extract preferences from web or query logs, analyze the browser history and derive interest weights from the numbers of clicks or the time spent on a page. According to findings in *Gauch et al. (2007)*, there is no significant evidence that an explicit user feedback mechanism results in better personalized content than implicitly recorded user information. Therefore, personalized applications nowadays mainly employ implicit profiling techniques, since they are less intrusive from a user's perspective.

In the context of scholarly applications, user profiles have been used in ad-hoc approaches, such as the expertise retrieval system used at Tilburg University (UvT) (https://www.tilburguniversity.edu/), academic search engines like AMiner (https://aminer.org) or personalized paper recommendations in Google Scholar (https://scholar.google.com).

The most dominant representation of user characteristics in this type of application is a weighted vector of keywords. This simple mathematical description permits classical information filtering algorithms, such as cosine similarity (*Manning, Raghavan & Schütze, 2008*), in order to measure item-to-item, user-to-user and item-to-user similarity.

### Competence management

E-learning applications were the earliest systems to provide personalized content based on a user's background knowledge and skill sets. The identified competence gaps were used to find appropriate learning strategies to gain the required knowledge to pass a course or to fulfill a certain task.

According to the definition in *HR-XML-Consortium (2004)*, a competency is "*A specific, identifiable, definable, and measurable knowledge, skill, ability and/or other deployment-related characteristic (e.g., attitude, behavior, physical ability) which a human resource may possess and which is necessary for, or material to, the performance of an activity within a specific business context*". *Draganidis & Mentzas (2006)* further analyzed the term *competency* and outlined four dimensions a competence can be described along: *category* (generic term for a group of similar skills), *competency* (the description of the competence term), *definition* (user scenarios that illustrate this competence) and *demonstrated behaviour* (explanations that clarify if the desired competency has been achieved).

The terms *competence* and *competency* are usually used synonymously. However, *Teodorescu (2006)* argues that there is a subtle difference in their meaning: while *competency* is mainly focused on the description of skills a person is supposed to possess in order to achieve a certain target, the term *competence* actually points to the measurement of skills to determine a certain level of expertise. While we appreciate this distinction, we consider the terms as synonymous in this article, but for the sake of consistency use the term *competence*.

### Semantic publishing

The emerging research domain of *Semantic Publishing* aims at making scientific knowledge accessible to both humans and machines. Semantic technologies have become increasingly important in the management of research objects. They enable automated systems to understand the meaning (semantics) and infer additional knowledge from published documents and data (*Berners-Lee & Hendler, 2001*; *Shadbolt, Hall & Berners-Lee, 2006*). Essential building blocks for the creation of structured, meaningful web content are information extraction and semantic *annotations*. These annotations are added to documents using special markups with predefined meanings to explicitly mark their structure (e.g., different sections of an article) and semantics (e.g., a publication's contributions, methods, or application domains) (*Sateli & Witte, 2015*). The semantically annotated documents can then be used in a multitude of applications, for example, in information retrieval systems, by finding specific document sections or measuring the similarity of different articles' contributions. In recent years, the semantic publishing community increasingly built and adopted controlled vocabularies, based on Semantic Web technologies, for describing research objects. However, despite the promises of better knowledge access (*Berners-Lee & Hendler, 2001*; *Shotton, 2009*), the manual annotation of existing research literature is prohibitively expensive for a wide-spread adoption. Consequently, a significant focus of research in the semantic publishing domain is dedicated to finding automatic approaches to annotate the wealth of existing scientific literature with shared vocabularies, based on approaches from the natural language processing and text mining domains.

## LITERATURE REVIEW

We focus our review on two core aspects: firstly, existing semantic vocabularies that describe scholars in academic institutions with their publications and competences, in

order to establish semantic user profiles. And secondly, we examine existing approaches for automatic profile generation through NLP methods.

## Vocabularies for scholarly user modeling

In the area of user modeling, a multitude of semantic approaches have emerged in the last decade that go beyond representing users interests with keywords in favour of using concepts of domain ontologies, for example in a vector-based model (*Sieg, Mobasher & Burke, 2007*; *Cantador & Castells, 2011*). In addition to providing a common understanding of domain knowledge, using semantic technologies also fosters the evolution towards more generic user models. An important goal of generic user modeling is facilitating software development and promoting reusability (*Kobsa, 2001*) of profiles across applications or platforms. Semantic web technologies, such as the representation of user characteristics in an RDF or Web Ontology Language (OWL) (https://www.w3.org/OWL/) compliant format can leverage this idea. In the following section, we review different proposals for generic user and competence modeling with semantic web vocabularies. Furthermore, we discuss scholarly ontologies that describe users, institutions and publications in the scientific domain.

### Vocabularies for competence modeling

From *Paquette (2007)*'s perspective, competences are phrases that connect a user's skills and positions to knowledge. This idea is reflected in his proposed competence ontology containing five main concepts, namely, *competence statement*, *competence*, *skill*, *knowledge entity* and *resource*. Paquette further developed his ontology into sub-ontologies for skills and performance indicators that could be incorporated in ontology-driven e-learning systems. However, Paquette's ontology was designed to be used within his proposed software framework and is essentially missing connections to other existing ontologies.

Another ontology approach is proposed by *Fazel-Zarandi & Fox (2012)*. Based on the assumption that someone who possesses appropriate skills is able to perform certain tasks, the ontology models skills at a certain level of proficiency and permits inter-linking with activities and knowledge fields.

The IntelLEO Competence Ontology (*Jovanovic et al., 2011*) aggregates the findings from different competence ontologies (*Schmidt & Kunzmann, 2006*; *Sandberg, 2000*; *Sitthisak, Gilbert & Davis, 2009*; *Paquette, 2003*; *Sicilia, 2005*; *Sampson & Fytros, 2008*) and defines a competence with respect to the corresponding domain-topic, skill and competence level. The IntelLEO ontology permits defining a competence as a prerequisite for another competence and provides the vocabulary for describing the process of gaining that specific skill. A competence *record*, hence, comprises a certain competence level, the source where the acquired competence has been achieved, as well as the date and methods that have been utilized to verify the acquisition.

### Vocabularies for semantic user profiles

GUMO (*Heckmann et al., 2005*) was the first generic user model approach, designed as a top-level ontology for universal use. This OWL-based ontology focuses on describing a user in a situational context, offering several classes for modeling a user's

personality, characteristics and interests. Background knowledge and competences are considered only to a small degree. In contrast, the IntelLEO (http://intelleo.eu/) ontology framework is strongly focused on personalization and enables describing preferences, tasks and interests. The IntelLEO framework consists of multiple RDFS-based ontologies, including vocabularies for user and team modeling, as well as competences as we described in 'Vocabularies for competence modeling'. The IntelLEO vocabularies are inter-linked with other user model ontologies, such as Friend of a Friend (FOAF) (http://www.foaf-project.org/). Thanks to the simplicity and inter-linking of FOAF to other Linked Open Vocabularies (LOVs), it has become very popular in recent years and is integrated in numerous personalized applications (*Celma, 2006*; *Raad, Chbeir & Dipanda, 2010*; *Orlandi, Breslin & Passant, 2012*). FOAF's RDF-based vocabulary provides for describing basic user information with predefined entities, such as name, email, homepage, and interests, as well as modeling both individuals and groups in social networks. However, FOAF does not provide comprehensive classes for describing preferences and competences, so that it would become directly usable within a scholarly application context.

Other ontologies aiming to unify user modeling vocabularies in semantic web applications are the Scrutable User Modeling Infrastructure (SUMI) (*Kyriacou, Davis & Tiropanis, 2009*) and the ontology developed by *Golemati et al. (2007)*. Besides general user information such as contact, address, preferences, education and profession, (*Golemati et al., 2007*) also provides a vocabulary for a user's activities in a given timeline. In contrast, SUMI (*Kyriacou, Davis & Tiropanis, 2009*) models user interests from the profiling perspective, which can be either explicitly given by the user or implicitly recorded by the system. The user model in SUMI is divided into four categories. The first two categories contain the manually provided user information: (i) generic personal user data and (ii) interests that are only specific for a certain application, e.g., preferences that are only applicable within the 'Amazon.com' platform. SUMI automatically stores the recorded user preferences in two further categories: (i) generic application information that are valid across different service providers and (ii) application-specific data that is only used for a certain service. Neither SUMI nor Golemati's approach are inter-linked with other vocabularies, and they are also not maintained anymore.

### Vocabularies for modeling scholars

For modeling scholars in the scientific domain, VIVO (http://vivoweb.org/ontology/core#) (*Börner et al., 2012*) is the most prominent approach that has been used in numerous applications (http://duraspace.org/registry/vivo). It is an open source suite of web applications and ontologies used to model scholarly activities across an academic institution. However, VIVO offers no support for content adaptation, due to missing classes for user interests, preferences and competences.

Another vocabulary modeling scientists and publications in research communities is the Semantic Web for Research Communities (SWRC) (http://ontoware.org/swrc/), which was developed to provide an ontological infrastructure for semantic information portals (*Gonzalez & Stumme, 2002*; *Haase et al., 2006*). The Semantic Web Portal Ontology (SWPO) (http://sw-portal.deri.org/ontologies/swportal) uses the FOAF, BIBTEX

**Table 1** **Comparison of existing user model and scholarly vocabularies:** *High* denotes that the vocabulary provides numerous classes and properties for the description of that entity. *Medium* means several classes and properties are available to define that entity and *Low* states that there is only one class and property available. *n/a* indicate that this entity is not covered by the vocabulary.

| Name | Coverage | | | | | | | |
|---|---|---|---|---|---|---|---|---|
| | Scientist | Role | Document | Research object | Project | Competence | Task | Interest |
| VIVO | High | High | High | Medium | Low | *n/a* | *n/a* | Low |
| LSC | Low | *n/a* | Medium | Low | Low | *n/a* | *n/a* | Low |
| SWPO | Medium | *n/a* | Medium | *n/a* | Low | *n/a* | Low | Low |
| SWRC | Medium | Medium | Medium | *n/a* | Low | *n/a* | *n/a* | *n/a* |
| AIISO | Medium | High | *n/a* | *n/a* | Low | *n/a* | *n/a* | Low |
| FOAF | Low | *n/a* | Low | *n/a* | Low | *n/a* | *n/a* | Low |
| GUMO | Low | Low | *n/a* | *n/a* | *n/a* | Medium | *n/a* | High |
| IntelLEO | Low | Low | Low | *n/a* | Low | High | Medium | Medium |

(http://purl.org/net/nknouf/ns/bibtex), Time (http://www.wsmo.org/ontologies/dateTime#) and DCMI Metadata (http://purl.org/dc/elements/1.1/) vocabularies to model researchers with their personal and scholarly activities, like publications and conferences. The goal of SWPO was to design a vocabulary to use within a portal application, where researchers working within a common scientific area can communicate with each other. The Linked Science Core Vocabulary (LSC) (http://linkedscience.org/lsc/ns#) provides a set of vocabularies for describing and interlinking researchers, publications and scientific data. LSC is very limited in its expressiveness and uses a small number of classes for describing research rhetoric elements in publications (e.g., hypothesis, data, method), rather than modeling researchers and their context. AIISO, the Academic Institution Internal Structure Ontology (http://purl.org/vocab/aiiso/schema), is a linked open vocabulary for describing the roles people play within various internal entities of an academic institution through its integration with FOAF and *participation vocabularies.*

One of the use cases we had in mind when designing our *ScholarLens* methodology was its application within sophisticated information filtering systems for scholars that consider a user's research background. Therefore, we explored the generic user models and scholarly ontologies reviewed above, in order to determine how well they can express features of scientific user modeling. The outcome of our study is summarized in Table 1: *Scientist* describes how well general user information about a scholar including name, address, affiliation, department and contact details can be defined. The ability to represent a user's roles, e.g., a student, a post-doc or a professor, is expressed with *Role. Document* refers to all kinds of scholarly publications and *Research Object* comprises the possibility to define all different types of data used or produced in scientific workflows. A user might be involved in different research projects, which is addressed with the concept *Project. Competence* points to the possibility of characterizing a user's expertise; whereas *Task* covers how well a user's responsibilities can be described. *Interest* refers to a user's preferences and interests for certain topics.

A comprehensive description of an academic user is provided by VIVO. This ontology is widely used in applications at academic institutions for representing general user

information and specific research areas and background. However, it does not provide for personalization, due to missing classes for user interests and preferences. In contrast to VIVO, the IntelLEO ontology framework is strongly focused on personalization and offers several classes for preferences, tasks and competences. The most prominent, but also most basic user model ontology is FOAF, which provides only very few classes and properties for user modeling.

## Implicit profile generation

Generic user modeling requires new methods for user profiling. Merely observing a user's browsing behavior is not enough for various tasks a scholar is involved in. More complex user information can be obtained from, e.g., context resources, such as affiliations a scholar is associated with, but also from content sources, for instance, a user's publications. Utilizing NLP techniques in user modeling has quite a long history (*Zukerman & Litman, 2001*) and has also become important in recent years in Information Retrieval (IR) for extracting named entities from scientific papers in order to profile scholars and to find an expert in a certain topic. Therefore, in the following subsections we focus on related work in expert profiling based on publications. Since social media offers an abundance of user information, we also report on implicit profiling approaches using data from social networks that are targeted at general users, rather than only scholars.

### *Expert profiling in information retrieval*

Certain tasks in research require finding experts with adequate background knowledge and skills. This so-called *Expertise Retrieval* gained a lot of attention in the Information Retrieval community (*Balog et al., 2012*), in particular with the expert finding task at the Enterprise Track of the *NIST (2009)* competition, which leveraged the idea of finding the right person with proper competences. Based on a corpus of different kinds of documents, such as web pages, source code files and emails from mailing-lists, the task was to return a ranked list of persons with appropriate background knowledge in a given domain.

However, as *Balog & De Rijke (2007)* point out, just matching people and domains in isolation is not enough: expert seekers often want to retrieve context information about a scholar's research network. The information with whom a person corresponds or collaborates might provide evidence on his or her establishment in the research community. Thus, it is getting more and more important to create comprehensive expert profiles, rather than just finding experts in documents for a given topic. Balog and Rijke clearly distinguish between *expert finding* and *expert profiling*. According to their definition, expert finding addresses the problem of finding experts for certain topics, based on a given set of documents, while expert profiling aims at creating profiles of individuals. The main goal in expert profiling is to establish a description of a person's competences and his or her social network. They divide the task of expert profiling into two stages: extracting topics and determining a person's competence in that topic. Here, competence is modeled as an association between query terms (topics) and candidate experts, where associations are mainly established based on textual evidence. A simple association would be the authorship of publications, where the content of the papers are the textual evidence of the candidate's

expertise: "*The stronger the association between a person and a topic, the likelier it is that the person is an expert on that topic*" (*Balog et al., 2012*). Different approaches have emerged in expertise retrieval for modeling these associations. According to *Balog et al. (2012)*, they can be grouped into five categories: *generative probabilistic models*, such as language or topic models, which determine the likelihood that a given topic is linked to a person, *discriminative models* computing the conditional probability that a tuple of topic and candidate expert is relevant, *voting models* that describe the generation of associations as a voting process, where scores from different sources are combined, *graph-based models* that analyze relationships among people and documents and describe associations along nodes (people and documents) and directed edges (conditions), as well as *other models* covering the broad spectrum of other approaches for modeling associations.

As *Balog et al. (2012)* point out, comparing all of the above mentioned approaches is a difficult task, since they are all influenced by different variables and components. Furthermore, they are often built into larger applications, where only the final system was evaluated. Therefore, *Balog et al. (2012)* only analyzed the range of the best score from the main test collections: W3C (http://research.microsoft.com/en-us/um/people/nickcr/w3c-summary.html) and CERC (http://es.csiro.au/cerc/) (both from TREC) and the UvT Expert Collection (http://ilps.science.uva.nl/resources/tu-expert-collection/) which is a collection of public data from about 1,168 experts from UvT. The Mean Average Precision (MAP) scores in the W3C collection varies between 0.20 and 0.30 on a query set from 2005 and between 0.45 and 0.65 on the dataset from the 2006 TREC competition. It needs to be considered that in this TREC collection, no actual 'relevance assessment' has been made, as the membership of the W3C people in different working groups was used to derive that a person has expertise in a certain topic. The MAP values from the CERC collection range from 0.45 to 0.60, whereas the MAP values for the expert profiling task on the UvT collection varies between 0.20 and 0.30. In the UvT collection, the ground truth was explicitly given by the users, as they provide a description of their research areas together with keywords from a topic hierarchy.

Another notable example for an expertise retrieval system is AMiner (https://aminer.org) (*Tang et al., 2010*), a system that combines user profiling and document retrieval techniques. General user information, such as affiliation and position, as well as research interests and research networks are presented in textual and visual form. The profiling approach consists of three main steps: profile extraction, author name disambiguation and user interest discovery. *Profile extraction* points to collecting general user information from web pages. Given a scholar's name, a binary classifier selects web pages according to features, like a person's name appearing in a page title. All retrieved pages are tagged with categories that are used to generate profile properties, including affiliation, email, address and phone numbers. Extracting research interests are left out in this step, since not all scholars enumerate their interests on their web pages. In addition, research interests should be proved by textual evidence. In a second step, AMiner attempts to link documents with the basic user profiles, in order to obtain a list of a scholar's publications. In this step, AMiner uses publications from different online digital libraries, e.g., DBLP or ACM. To solve the *name disambiguation* problem (i.e., two scholars with the same name), they

developed a probabilistic model based on author names. In the final step, they determine *user interests* from the generated linked list of papers. Interests are described based on the detected topics. A topic consists of a mixture of words and probabilities being associated with that word. They propose a probabilistic model called Author-Conference-Topic (ACT) model, where 'conference' comprises all kinds of publications, namely journals, conferences and articles. The idea behind this approach is that an author writing a paper uses different words based on her research interests, which denote the topic distribution. The discovered topic distributions are used as research interests and are stored together with the general information in an extended FOAF format, in what they call a researcher network knowledge base (RNKB). For the evaluation, they utilized pooled relevance judgments (*Buckley & Voorhees, 2004*) and human judgments. Seven people rated the retrieved expert lists for 44 topic queries along four expertise levels: definite expertise, expertise, marginal expertise and no expertise. The judges were taught to do the rating according to a guideline following certain criteria, such as how many publication the retrieved scholar actually has for the given topic or how many awards she has received or conferences attended. In a final step, the judgment scores were averaged. In their experiments, they tested different language models along with their ACT model, which was shown to outperform the other models in the best run (P@5: 65.7%, P@10: 45.7%, MAP: 71%).

### Expert profiling using text mining

Generating scholarly profiles has not only been investigated in Information Retrieval, but also in the computational linguistics domain. A first expert profiling approach using Linked Open Data is suggested by *Buitelaar & Eigner (2008)*. They define simple linguistic patterns to identify competences in a user's research publications. *Bordea & Buitelaar (2010b)* further developed that idea using a GATE pipeline (*Bordea & Buitelaar, 2010a*) that finds pre-defined skill types in research papers. They define skill types as general domain words that represent theoretical and practical expertise, such as *method*, *algorithm* or *analysis*. Additionally, they applied an adapted TD-IDF filtering algorithm and removed terms from the final list that were considered too broad. In *Bordea et al. (2012)*, they extended their system with semantic linking to DBpedia ontology concepts (http://wiki.dbpedia.org/services-resources/ontology) and attempt to find a corresponding concept in the Linked Open Data cloud for each extracted topic. For the evaluation, they conducted a user study with three domain experts, using their own corpus. The users were asked to judge a limited list of 100 ranked topics for a given domain. The list was divided into three sections, *top*, *middle* and *bottom*, and the judges classified the provided topics into *good*, *bad* or *undecided*. Finally, the Kappa statistic was applied to aggregate the three judgments. Overall, 80% of the top ranked topics were marked as *good*.

According to recent findings *Letierce et al. (2010)*, social media platforms are widely used among scientists to share research news. *Nishioka & Scherp (2016)* generated scholarly profiles out of social media items, namely Twitter (http://www.twitter.com), for recommending scientific publications. They examine different factors influencing the recommendation process, such as profiling method, temporal decay (sliding window and exponential decay) and richness of content (full-text and title versus title only). Regarding

the profiling method, they took into account the following filtering methods: CF-IDF, an adapted TF-IDF algorithm using concepts of ontologies instead of full-text terms, HCF-IDF, their own extended hierarchical approach and Latent Dirichlet Allocation (LDA) (*Blei, Ng & Jordan, 2003*) topic modeling. For both user tweets and publications, they extract concepts with corresponding labels in the underlying knowledge base through gazetteers. By means of the Stanford Core NLP (http://stanfordnlp.github.io/CoreNLP/) tools, they remove stop words and Twitter hashtags. In their evaluation with 123 participants and around 280,000 scientific publications from economics, they analyzed in total 12 different recommendation strategies, derived as combinations from the three influencing factors and their sub-factors. The participants obtained the top-5 recommendations for each of the 12 strategies and rated the presented publication list on a binary scale. Their results reveal that the most effective strategy was the one with the CF-IDF filtering, the sliding window, and with full-texts and titles. Additionally, it turned out that using titles only in combination with the HCF-IDF filtering produces similarly good recommendations.

### Implicit profiling in social media

In the last decade, using social media platforms for implicit user profile generation attracted increasing attention (*Szomszor et al., 2008*; *Abel et al., 2011*; *Stankovic, Rowe & Laublet, 2012*). Through a number of different NLP methods, 'interesting' topics are extracted from short messages or social network posts. LinkedVis (*Bostandjiev, O'Donovan & Höllerer, 2013*) for instance is an interactive recommender system that generates career recommendations and supports users in finding potentially interesting companies and specific roles. LinkedVis developers designed four different user models based on data from LinkedIn (https://www.linkedin.com) and extract interests and preferences from a user's connections, roles and companies. Two of the four constructed profiles contained meaningful entities instead of plain keywords. A Part-of-Speech tagger was utilized to find noun phrases that were mapped to Wikipedia articles. The evaluation with a leave-one-out cross-validation revealed that the user models with semantic enrichment produced more accurate and diverse recommendations than the profiles based on TF-IDF weights and occurrence matching.

Another approach using NLP methods for online profile resolution is proposed by *Cortis et al. (2013)*: they developed a system for analyzing user profiles from heterogenous online resources in order to aggregate them into one unique profile. For this task, they used GATE's ANNIE (https://gate.ac.uk/sale/tao/splitch6.html) plugin (*Cunningham et al., 2011*) and adapted its JAPE grammar rules to disassemble a person's name into five sub-entities, namely, prefix, suffix, first name, middle name and surname. In addition, a Large Knowledge Base (LKB) Gazetteer was incorporated to extract supplementary city and country values from DBpedia (http://dbpedia.org). In their approach, location-related attributes (e.g., 'Dublin' and 'Ireland') could be linked to each other based on these semantic extensions, where a string-matching approach would have failed. In their user evaluation, the participants were asked to assess their merged profile on a binary rating scale. More than 80% of the produced profile entries were marked as correct. The results

reveal that profile matchers can improve the management of one's personal information across different social networks and support recommendations of possibly interesting new contacts based on similar preferences.

## Discussion

As presented above, automatic user profiling approaches using linked named entities and NLP techniques are becoming increasingly popular. Sources for generating profiles vary from scientific papers to social media items, profiles in social networks or an aggregation of different sources. In particular, expert profiling has evolved into its own research area. However, the most widespread description of a user model in these applications is still a term-based vector representation. Even though keywords are increasingly replaced by linked entities, they still lack an underlying semantic model in RDF or OWL format.

In contrast, we aim at automatically creating semantic user profiles for scholars by means of NLP methods and semantic web technologies. Our goal is to establish user profiles in an interoperable RDF format that can be stored in a triplestore. Hosting user information in such a structured and meaningful semantic format facilitates data integration across different sources. Furthermore, expressive SPARQL queries and inferences can help to discover related preferences that are not explicitly stated in the profiles. The open, shared knowledge base constructed by our approach can then be accessed by a multitude of different scholarly applications.

## DESIGN

Modeling semantic scholarly profiles requires a formalization of the relations between authors and their competences in a semantically rich format. The three central concepts in our model are *researchers*, their *competence topics* and a set of scholarly *documents*. We hypothesize that authors of a scholarly publication (e.g., a journal article) are competent in topics mentioned in the article to various degrees. This way, for each author (i.e., researcher with at least one publication) we create a *semantic user profile*. With the ultimate goal of creating machine-readable, inter-operable profiles, we decided to use the W3C standard Resource Description Framework (RDF) to design profiles based on semantic triples.

### Semantic modeling of user competence records

Every entity in our model is defined as a semantic triple $\langle s, p, o \rangle$, where $s$ is a unique resource identifier (URI) within the knowledge base, $p$ is a property from a set of pre-defined relations between authors and their profile elements, and $o$ is either a unique identifier of an entity (in the knowledge base or an external dataset), or a literal value.

All users in our model are instances of the User Model Ontology (UM) (http://intelleo.eu/ontologies/user-model/spec/) User class, designed specifically for modeling users in collaborative contexts. As a subclass of the FOAF Person class, all user instances inherit attributes and relations from the FOAF vocabulary. Since our primary source of competence detection are the users' publications, we also need to semantically model the documents and their content. For a user $u$, we denote $\mathbf{D}_u = \{d_1, d_2, \ldots, d_n\}$ as a set of documents published by the user. Each document $d$ contains a set of topics

$\mathbf{T}_d = \{t_1, t_2, \ldots, t_n\}$, where each $t_i$ is essentially a named entity (NE), like '*text mining*' or '*automatic reasoning*' within the document's sentences. Topics are often repeated many times in a document. Thus, we need to be able to annotate where a topic is mentioned in a document. Every topic $t$ will then have two attributes: a *start* and *end* offset, containing the index of their first and last character in the document. This way, every mention of a competence topic found in a document is modeled as a semantic triple and we can count the frequency of the competence topics mentioned in each document, as well as overall for each user profile. The challenge here is to identify various orthographic forms of the same topic in a document, so that we can determine the unique topics in $T$. For instance, topic '*Linked Open Data*' may be mentioned in the abstract of a document and then later on '*LOD*' may appear in the background section. This results in two triples, each describing the two competence topics with different forms in text, although, in fact they represent the same concept. As a solution, each mention of topic $t$ will be the subject of an additional triple $\langle t, \text{rdfs}:\text{isDefinedBy}, l \rangle$, where $l$ is the URI of a resource, where the topic (concept) is defined in a given knowledge base or dataset. Unique topics with different surface forms in the document can now be identified through their resource URI.

In our model, users and their competence topics are inter-connected through *competence records*. A competence record contains the provenance metadata of a user's competences (e.g., the document identifier in which it was found) and can be additionally associated with a *level* of expertise. Finally, we define a *user profile* as a labeled, directed graph $\mathbf{P}_u = (V, E)$, where $\mathbf{V} = \{D \cup T\}$ and $\mathbf{E}$ is a set of edges between a user's publications and their encompassed topics, as well as outgoing links from the $T$ elements to LOD resources. Since RDF documents intrinsically represent labeled, directed graphs, the semantic profiles of scholars extracted from the documents can be merged through common competence URIs—in other words, authors extracted from otherwise disparate documents can be semantically related using their competence topics.

Following the best practices of producing linked open datasets, we tried to reuse existing Linked Open Vocabularies (LOVs) to the extent possible for modeling the semantic user profiles. Table 2 shows the vocabularies used to model our semantic scholar profiles and their respective selected terms. We largely reuse IntelLEO (http://www.intelleo.eu/) ontologies for competence modeling—originally designed for semantic modeling of learning contexts—, in particular the vocabularies for *User and Team Modeling* (http://intelleo.eu/ontologies/user-model/spec) and *Competence Management* (http://www.intelleo.eu/ontologies/competences/spec). We also reuse the PUBO ontology (*Sateli & Witte, 2015*) for modeling the relations between the documents that we process, the generated annotations and their inter-relationships. Figure 2 shows a minimal example semantic profile in form of an RDF graph.

## Automatic detection of competences

In this section, we describe our automatic workflow for constructing semantic scholarly profiles. We first iterate the requirements of such a workflow and then present various components of our *ScholarLens* approach that satisfy these requirements. An overview of our system architecture is illustrated in Fig. 3.

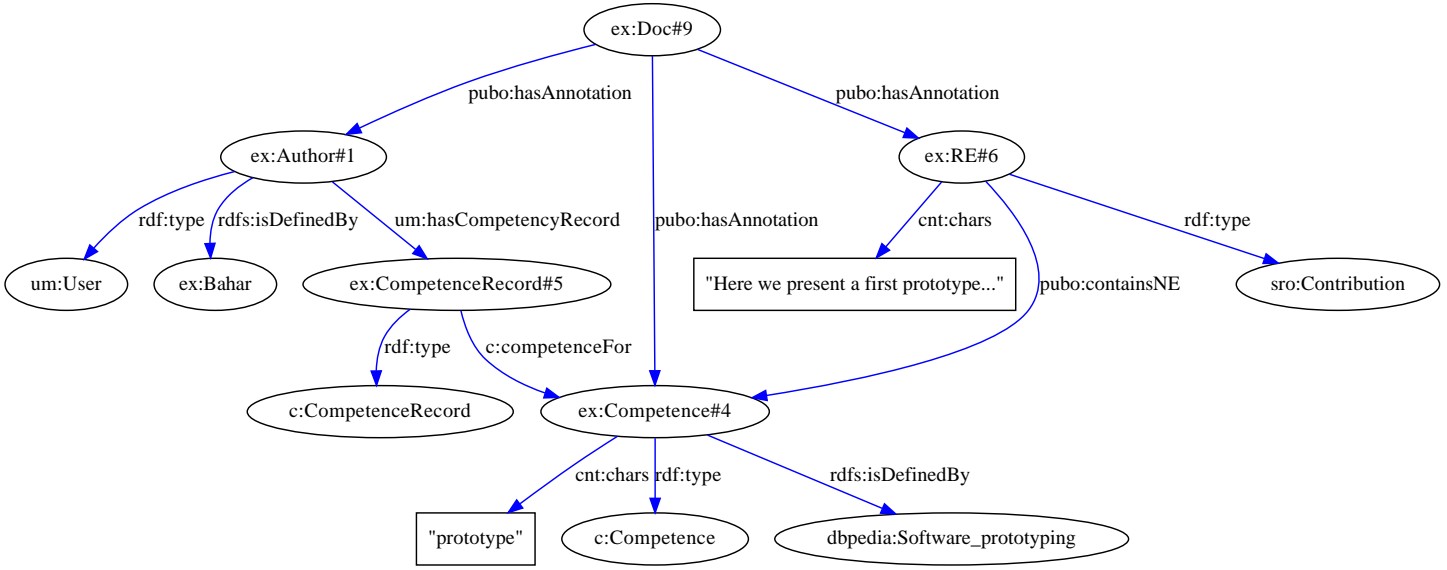

**Figure 2** The RDF graph shown in this picture represents a semantic user profile that illustrates the relations between an author and the topics mentioned in her article.

**Table 2** Modeling user profiles with linked open vocabularies (LOVs): this table shows the terms selected from various vocabularies and their corresponding concept in the semantic user profiles. The vocabulary namespaces used in the table can be de-referenced using the URLs shown at the bottom.

| LOV Term | Modeled concept |
|---|---|
| um: user | Scholarly users, who are the documents' authors. |
| um:hasCompetencyRecord | A property to keep track of a user's competence (level, source, etc.). |
| c: competency | Extracted topics (LOD resources) from documents. |
| c:competenceFor | A relation between a competence record and the competence topic. |
| sro: rhetoricalElement | A sentence containing a rhetorical entity, e.g., a *contribution*. |
| cnt:chars | A competence's label (surface form) as it appeared in a document. |
| pubo:hasAnnotation | A property to relate annotations to documents. |
| pubo:containsNE | A property to relate rhetorical zones and entities in a document. |
| oa:start & oa:end | A property to show the start/end offsets of competences in a text. |

**Notes.**
um, http://intelleo.eu/ontologies/user-model/ns/; c, http://intelleo.eu/ontologies/competences/ns/; sro, http://salt.semanticauthoring.org/ontologies/sro#; cnt, http://www.w3.org/2011/content#; pubo, http://lod.semanticsoftware.info/pubo/pubo#; oa, http://www.w3.org/ns/oa/; rdf, http://www.w3.org/1999/02/22-rdf-syntax-ns#; rdfs, http://www.w3.org/2000/01/rdf-schema#.

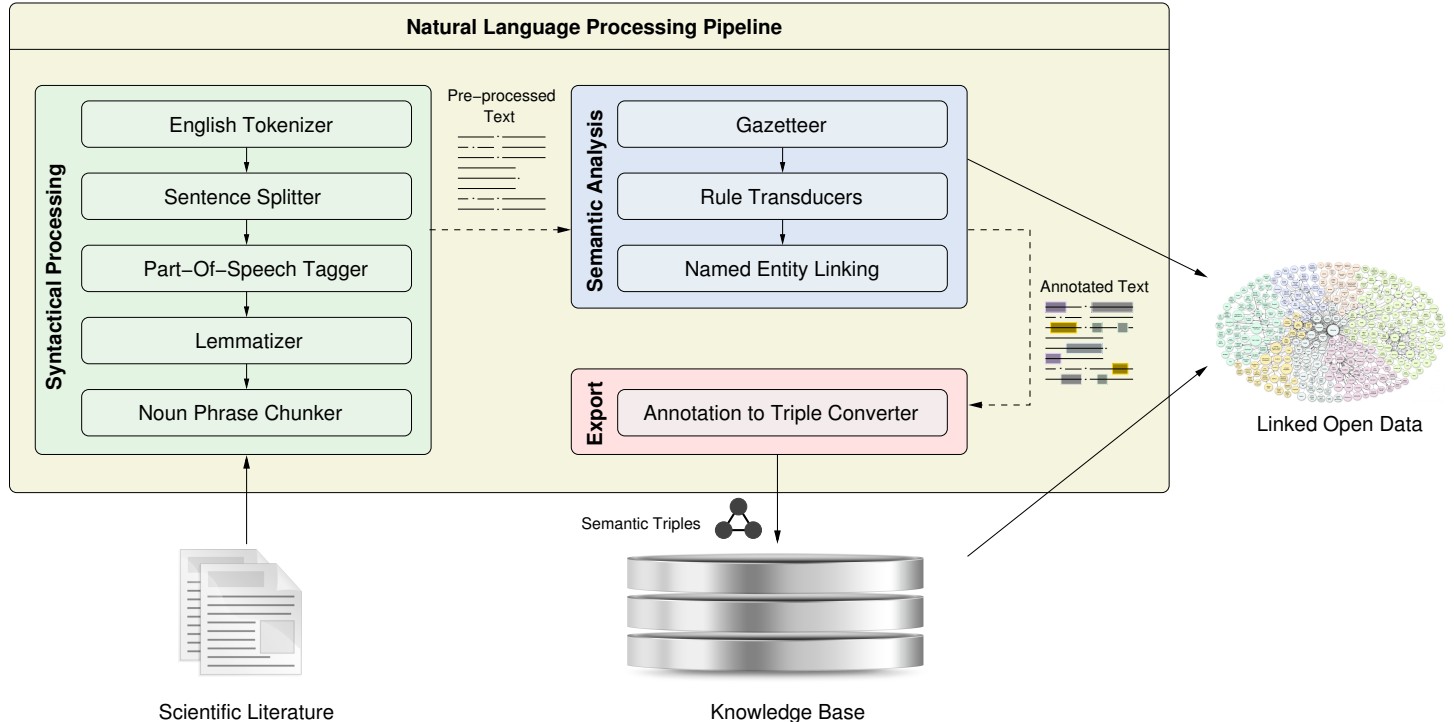

**Figure 3** **The workflow shown here depicts how scientific literature undergoes various syntactical and semantic processing steps.** The output of the workflow is a knowledge base populated with semantic user profiles, inter-linked with resources on the linked open data cloud.

### Requirements

Our goal is to automatically populate semantic user profiles by mining the users' publications for competence topics. Therefore, we identify the following requirements for our workflow:

*Requirement 1: access to scholarly articles' full-text.* The workflow should be able to accept a set of documents written by an author as input, which may be in various publisher-dependent, formatting styles. The documents must be machine-readable, that is, the workflow must be have access to the textual content of the entire article.

*Requirement 2: automatic extraction of domain topics.* In order to extract the competence topics, we need to annotate the Named Entities (NEs) in a document that represent relevant concepts for a domain. For example, words like '*benchmarking*' and '*Linear Regression*' represent relevant research activities and concepts in a computer science article.

*Requirement 3: semantic representation of extracted information.* The extracted information from documents must be stored in a machine-readable and inter-operable format, in order to facilitate the implementation of value-added services, like expertise recommendation. The users, their publications and the competence topics must be uniquely identifiable in the output. All instances and their attributes must be represented using semantic web vocabularies.

### Text mining documents for competence topics

We leverage various text mining techniques in our semantic profiling workflow. If the input document is in any format other than plain-text, it first goes through a text extraction phase. In this step, any publisher-specific formatting is eliminated from the document. The plain-text document is then prepared for further analysis in a so-called *pre-processing* phase. This step is language-dependant, but can be reused for documents in various domains. The first step is to break down a document's text into *tokens*—smaller, linguistically meaningful segments, like words, numbers and symbols. Tokens are then grouped into sentences and each word token is tagged with a part-of-speech tag, e.g., 'noun' or 'verb'. To eliminate the different orthographical forms of tokens in a text, we *lemmatize* the document's text, so that all inflected forms of nouns (singular vs. plural) and verbs (various tenses) are changed to their canonical form (Requirement 1). The pre-processed text is subsequently passed onto the *semantic processing* phase for user competence detection.

### Grounding competence topics to LOD resources

Since it is not feasible to manually construct and maintain a knowledge base of all possible topics appearing in documents, we leverage the Linked Open Data (LOD) cloud as a source of continually-updated knowledge. The idea is to link the competence topics to their corresponding resources on the LOD cloud, where machine-readable, semantic metadata about each topic can be found by de-referencing the link's address. To this end, we use a linked data-enabled Named Entity Recognition (NER) tool that can detect named entities in the documents, resolve them to their correct sense and link the surface forms to existing resources in LOD datasets (Requirement 2).

Grammatical processing performed in the previous step helps us to filter out tokens that do not typically represent competences, like adverbs or pronouns. We exclude processing the sentences in figure and table captions, formulas, section headers and references, as we empirically verified that these document regions rarely contain authors competence topics.

## Knowledge base population

As we mentioned earlier, the output of our system is a knowledge base populated with semantic user profiles (Requirement 3). We leverage the Resource Description Framework (RDF) syntax to describe the extracted information in a semantically meaningful way, using the model described in 'Semantic modeling of user competence records'. In the populated user profiles, we use the raw frequency of the detected topics (named entities) in documents as a means of ranking the top competence topics for each scholar. We store the profile RDF documents in a triplestore that can later on be queried for various applications.

An important design decision in modeling the knowledge base is deciding whether all (potentially hundreds) of topics in a document are indeed representative of its authors' competences. Or perhaps, a subset of the topics located in the *rhetorical zones* of an article are better candidates? Previously in *Sateli & Witte (2015)*, we investigated the detection of *Rhetorical Entities* (REs) as sentences in a scholarly document that convey its authors findings or argumentations, like their Claims or Contributions. We also showed how the named entities within the boundaries of these REs can represent a document's content

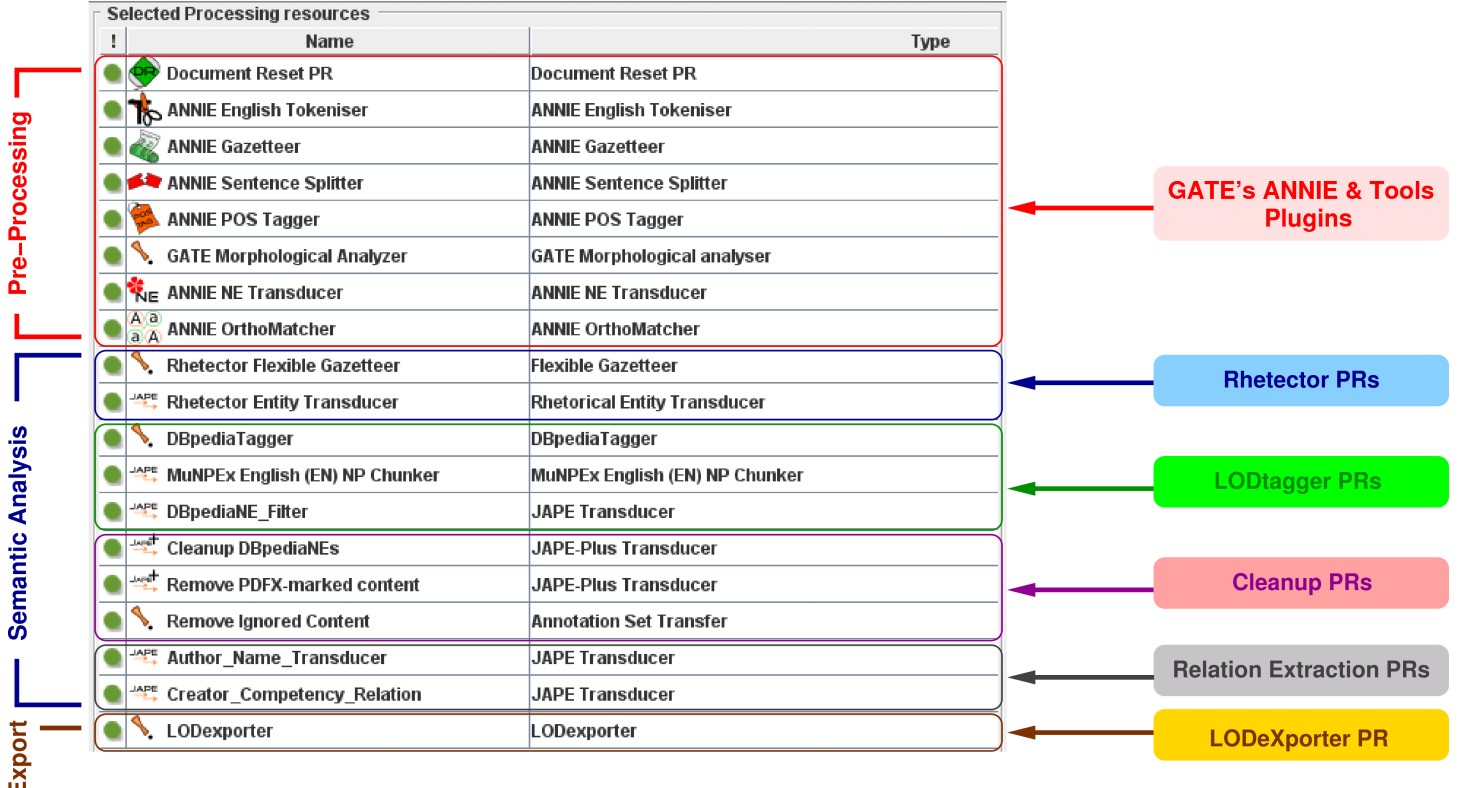

**Figure 4** This figure shows a sequence of processing resources aggregated into a GATE pipeline for the automatic generation of user profiles based on competences extracted from publications.

in various use cases, such as retrieving semantically related articles. Additionally, we determined that storing the named entities within REs requires an order of a magnitude fewer triples, compared to exporting the topics of the entire document. To test whether the same assumption can be made about the authors' competence topics, we additionally export all rhetorical entities in a document and add an additional property to each topic (NE) that is mentioned within an RE. We will revisit our hypothesis in 'Extended experiments'.

## IMPLEMENTATION

In this section, we describe how we realized the semantic user profiling of authors illustrated in the previous section.

### Extraction of user competences with text mining

We developed a text mining pipeline (Fig. 4), implemented based on the GATE framework (*Cunningham et al., 2011*), to analyze a given author's papers in order to automatically extract the competence records and topics. The NLP pipeline accepts a corpus (set of documents) for each author as input. We use GATE's ANNIE plugin to pre-process each document's full-text and further process all sentences with the Hepple part-of-speech (POS) tagger, so that their constituents are labeled with a POS tag, such as *noun*, *verb*, or *adjective* and lemmatized to their canonical (root) form. We then use MuNPEx

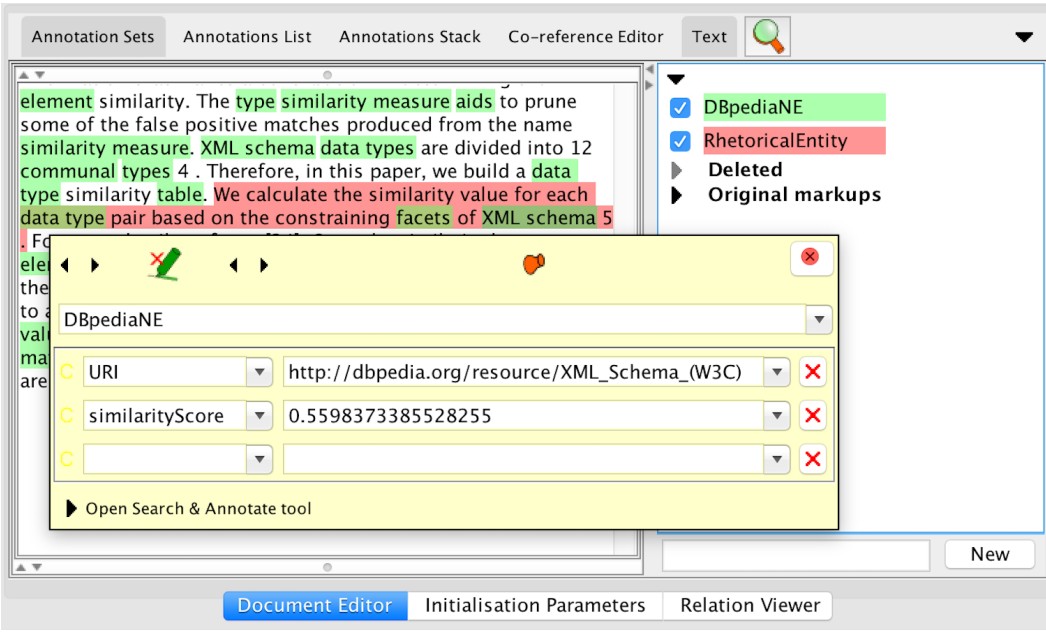

**Figure 5** Example annotated document for a contribution (RhetoricalEntity) and the competence topics (DBpediaNEs) within its boundaries in GATE graphical user interface.

(http://www.semanticsoftware.info/munpex), a GATE plugin to detect noun phrases in text, which helps us to extract competence topics that are noun phrases rather than nouns alone. Subsequently, we use our LODtagger (http://www.semanticsoftware.info/lodtagger), which is a GATE plugin that acts as a wrapper for the annotation of documents with Named Entity Recognition tools. In our experiments, we use a local installation of DBpedia Spotlight (*Mendes et al., 2011*) v7.0 with the statistical model version 20160113 (http://spotlight.sztaki.hu/downloads/) for English (en_2+2) (*Daiber et al., 2013*). Spotlight matches the surface form of the document's tokens against the DBpedia ontology and links them to their corresponding resource URI. LODtagger then transforms the Spotlight response to GATE annotations using the entities' offsets in text and keeps their URI in the annotation's features (Fig. 5).

To evaluate whether our hypothesis that the NEs within rhetorical zones of a document are more representative of the author's competences than the NEs that appear anywhere in a document, we decided to annotate the *Claim* and *Contribution* sentences of the documents using our Rhetector (http://www.semanticsoftware.info/rhetector) GATE plugin (*Sateli & Witte, 2015*). This way, we can create user profiles exclusively from the competence topics that appear within these RE annotations for comparison against profiles populated from full-text. Rhetector was evaluated in *Sateli & Witte (2015)* with an average F-measure of 73%.

Finally, we create a competence record between the author and each of the detected competences (represented as DBpedia NEs). We use GATE's JAPE language that allows us to execute regular expressions over documents' annotations by internally transforming them into finite-state machines. Thereby, we create a competence record (essentially,

a GATE relation) between the author annotation and every competence topic in the document. Figure 5 shows example semantic annotations generated by our text mining pipeline in the GATE Developer environment.

## Automatic population of semantic user profiles

The last step in our automatic generation of semantic user profiles is to export all of the GATE annotations and relations from the syntactic and semantic processing phases into semantic triples, represented with RDF. Our LODeXporter (http://www.semanticsoftware.info/lodexporter) tool provides a flexible mapping of GATE annotations to RDF triples with user-defined transformation rules. For example, the rules:

```
map:GATECompetence map:GATEtype "DBpediaNE" .
map:GATECompetence map:hasMapping map:GATELODRefFeatureMapping .
map:GATELODRefFeatureMapping map:GATEfeature "URI" .
map:GATELODRefFeatureMapping map:type  rdfs:isDefinedBy  .
```

describe that all '*DBpediaNE*' annotations in the document should be exported, and for each annotation the value of its '*URI*' feature can be used as the object of the triple, using '*rdfs:isDefinedBy*' as the predicate. Similarly, we use the LOV terms shown in Table 2 to model authors, competence records and topics as semantic triples and store the results in an Apache TDB-based (http://jena.apache.org/documentation/tdb/) triplestore.

## EVALUATION

We performed two rounds of evaluations: In a first user study, described in detail in *Sateli et al. (2016)* and summarized below, we tested an initial version of our profiling approach. To investigate the reasons for generated 'irrelevant' profile entries, we went back to a number of users from our study and performed a post-mortem error analysis. Our finding are presented in 'User study: error analysis'.

Based on the detected issues, we refined both our profile generation pipeline and the survey methodology for a second user study. Our new setup and the results are described in 'Extended Experiments'.

### Methodology and Metrics

We evaluate the quality of the generated user profiles through user studies. For each participant, we processed a number of the user's publications and created competence entries in a knowledge base. We then went back to the users and showed them the top-50 (by competence frequency) generated profile entries in a human-readable format, in order to evaluate whether we found a pertinent competence.

To evaluate the effectiveness of our system, we utilize common retrieval evaluation methods, namely *Precision@k*, *Mean Average Precision (MAP)* (*Manning, Raghavan & Schütze, 2008*) and *normalized Discounted Cumulative Gain (nDCG)* (*Järvelin & Kekäläinen, 2002*).

We first analyze the top ranked competence results of an individual user, specifically, the top-10, top-25 and top-50, and measure the precision at rank (Precision@k) Eq. (1),

which is defined as:

$$\text{Precision@k} = \frac{1}{k} \cdot \sum_{c=1}^{k} rel(c), \tag{1}$$

where $k$ denotes the rank of the competence that is considered and $rel(c)$ marks the rating for the iterating position $c$, which is either 0 for *irrelevant* or 1 for *relevant* topics. While the Precision@k is focused on the result for a certain rank of an individual user, the MAP is a metric that expresses the mean average of competence rankings over all users in one value. MAP Eq. (3) indicates how precise an algorithm or system ranks its top-k results, assuming that the entries listed on top are more relevant for the information seeker than the lower ranked results. Precision is then evaluated at a given cut-off rank $k$, considering only the top-$k$ results returned by the system. Hence, MAP is the mean of the average precisions at each cut-off rank and represents a measure for computing the quality of a system across several information needs; in our case, users with competences. For all relevant competences $c \in C$ per user $u$, we compute the Average Precision (AP) of a user $u$ as follows Eq. (2):

$$AP(u) = \frac{1}{C_{r,k}} \sum_{c=1}^{k} \text{Precision@k} \cdot rel(c), \tag{2}$$

where $rel(c)$ is 1 if the competence $c$ is relevant and 0 for the opposite case. $C_{r,k}$ is the set of all relevant competences up to a certain cut-off rank $k$. Finally, for every user $u \in U$, the MAP is then defined as follows Eq. (3):

$$\text{MAP}(U) = \frac{1}{|U|} \sum_{u=1}^{|U|} AP(u). \tag{3}$$

In contrast to the MAP, which only considers binary ratings, the DCG Eq. (4) computes the ranking based on Likert scales (*Likert, 1932*). Given a list of competences, $rel_c$ is the actual rating of each single competence $c$. For example, in our second user study we assign the competence types 0 (irrelevant), 1 (general), 2 (technical) and 3 (research), as defined below. Similar to the precision, the DCG assumes that higher ranked items are more relevant for the users than lower ranked. In order to take this into account, a logarithmic decay function is applied to the competences, known as the *gains*.

For a set of users $U$, let $rel(c)$ be the relevance score given to competence $c \in C$ for user $u \in U$. Then, the DCG for every user as defined in *Croft, Metzler & Strohman (2009)* is the sum over all $|C|$ competence gains Eq. (4):

$$\text{DCG}_u = rel_1 + \sum_{c=2}^{|C|} \frac{rel_c}{\log_2 c}. \tag{4}$$

Due to the variable length of the result lists, the DCG values should be normalized across all users. Therefore, the competences are sorted according to their relevance. This ordered list is called *Ideal DCG* (IDCG). Finally, this IDCG is used to compute the *normalized DCG* (nDCG) for a user $u$ as follows Eq. (5):

$$\text{nDCG}_u = \frac{\text{DCG}_u}{\text{IDCG}_u}. \tag{5}$$

## Initial experiments

In our first evaluation round (*Sateli et al., 2016*), we reached out to ten computer scientists from Concordia University, Canada and the University of Jena, Germany (this study also included the authors of the paper) and asked them to provide us with a number of their selected publications. We processed the documents and populated a knowledge base with the researchers' profiles. Using a Java command-line tool that queries this knowledge base, we generated LaTeX documents as a human-readable format of the researchers' profiles, each listing the top-50 competence topics, sorted by their occurrence in the users' publications. Subsequently, we asked the researchers to review their profiles across two dimensions: *(i)* relevance of the extracted competences and *(ii)* their level of expertise for each extracted competence. For each participant, we exported two versions of their profile: *(i)* a version with a list of competences extracted from their papers' full-text, and *(ii)* a second version that only lists the competences extracted from the rhetorical zones of the documents, in order to test our hypothesis described in 'Knowledge base population'. To ensure that none of the competence topics are ambiguous to the participants, our command-line tool also retrieves the English label and comment of each topic from the DBpedia ontology using its public SPARQL endpoint (http://dbpedia.org/sparql). The participants were instructed to choose only one level of expertise among ("*Novice*", "*Intermediate*", "*Advanced*") for each competence and select "*Irrelevant*" if the competence topic was incorrect or grounded to a wrong sense.

Table 3 shows the evaluation results of our first user study. In this study, a competence was considered as relevant when it had been assigned to one of the three levels of expertise. For each participant, we measured the average precision of the generated profiles in both the full-text and RE-only versions. The results show that for both the top-10 and top-25 competences, 70–80% of the profiles generated from RE-only zones had a higher precision, increasing the system MAP up to 4% in each cut-off. In the top-50 column, we observed a slight decline in some of the profiles' average precision, which we believe to be a consequence of more irrelevant topics appearing in the profiles, although the MAP score stays almost the same for both versions.

## User study: error analysis

In order to refine our approach, we went back to the users from the first study to understand the root cause of competences they marked as *irrelevant*. We asked a number of users to classify each irrelevant competence into one of four error categories:

Type 1 (Wrong URI). The profile contains a wrong URI: this is typically caused by the linking tool assigning the wrong URI to a surface form; either because it picked the wrong sense among a number of alternatives or the correct sense does not exist in the knowledge base.

Type 2 (Empty description). As explained above, we retrieve the *comment* for each competence URI to make sure users understand their profile entries. In about 3% of profile entries, this automatic process failed, leading to an empty description, which was often marked as irrelevant. We identified three main causes for this: *(a)* a timeout in the SPARQL query to the public DBpedia endpoint; *(b)* missing comment entry in

**Table 3 Evaluation results for the generated user profiles in the first user study: this table shows the number of distinct competence topics extracted from the ten participants and the average precisions at 10, 25 and 50 cut-off ranks.** The last row (MAP) shows the mean average precision of the system at various cut-offs.

| Participant | #Docs | #Distinct competences | | Avg. precision@10 | | Avg. precision@25 | | Avg. precision@50 | |
|---|---|---|---|---|---|---|---|---|---|
| | | Full-text | REs only | Full-text | REs only | Full-text | REs only | Full-text | REs only |
| R1 | 8 | 2,718 | 293 | **0.91** | 0.80 | **0.84** | 0.74 | **0.80** | 0.69 |
| R2 | 7 | 2,096 | 386 | **0.95** | 0.91 | 0.90 | **0.92** | 0.87 | **0.91** |
| R3 | 6 | 1,200 | 76 | 0.96 | **0.99** | 0.93 | **0.95** | **0.92** | 0.88 |
| R4 | 5 | 1,240 | 149 | 0.92 | **0.92** | **0.86** | 0.81 | **0.77** | 0.75 |
| R5 | 4 | 1,510 | 152 | 0.84 | **0.99** | 0.87 | **0.90** | 0.82 | **0.82** |
| R6 | 6 | 1,638 | 166 | 0.93 | **1.0** | 0.90 | **0.97** | 0.88 | **0.89** |
| R7 | 3 | 1,006 | 66 | 0.70 | **0.96** | 0.74 | **0.89** | 0.79 | **0.86** |
| R8 | 8 | 2,751 | 457 | 0.96 | **1.0** | 0.92 | **1.0** | 0.92 | **0.99** |
| R9 | 9 | 2,391 | 227 | 0.67 | **0.73** | 0.62 | **0.70** | 0.56 | **0.65** |
| R10 | 5 | 1,908 | 176 | **0.96** | 0.91 | 0.79 | **0.80** | 0.69 | **0.70** |
| | | | MAP | 0.88 | **0.92** | 0.83 | **0.87** | 0.80 | **0.81** |

English for some resources in the online DBpedia; and the much rarer cause *(c)* where the URI generated by the linking tool was valid for an older version of DBpedia, but has meanwhile been removed.

Type 3 (User misunderstanding). Some users interpreted the task differently when it came to identifying their competences: rather than evaluating what they are generally competent in, they marked each entry that did not fall into their research fields as irrelevant. For example, a researcher working on web services marked 'HTTP (Protocol)' as irrelevant, since HTTP was not a research topic in itself, though the user clearly had knowledge about it.

Type 4: (Unspecific competence). Users often assigned irrelevant for competences that were deemed too broad or unspecific. The cause is very similar to Type 3, with the main difference that competences here were high-level concepts, like *System*, *Idea*, or *Methodology*, whereas Type 3 errors were assigned to technical terms, like *HTTP*, *Data Set*, or *User (computing)*.

The results of this analysis are summarized in Table 4. As can be seen, the majority of the errors (77% and 82%) are of Type 1. This is consistent with earlier observations we had about DBpedia Spotlight when applying it to research literature (*Sateli & Witte, 2015*). Modifying or retraining Spotlight itself was out of the scope of this work, but we addressed some common errors in our pipeline, as described below.

## Extended experiments

With the lessons learned from our first experiment, we enhanced our competence topic detection pipeline to remove the error types iterated in the previous section. In particular, to address Type 1 error, we excluded exporting entities with surface forms like "*figure*" or "*table*" from newly generated profiles, as these were consistently linked to irrelevant topics like "*figure painting*" or "*ficus*". To address Type 3 and Type 4 errors, we refined the

**Table 4  Error analysis of the irrelevant competence entries generated for the participants in the first user study: for each error type, the total numbers of irrelevant competences in the profile and its percentage (rounded) is shown.**

| | Error type | User | | | | | | | Average |
|---|---|---|---|---|---|---|---|---|---|
| | | R8 | R1 | R6 | R9 | R3 | R4 | R2 | |
| **Full-text profiles** | Type 1 | 4 | 7 | 10 | 7 | 6 | 16 | 7 | 8.14 |
| | | 100% | 64% | 91% | 30% | 100% | 89% | 100% | 82% |
| | Type 2 | 0 | 2 | 0 | 1 | 0 | 0 | 0 | 0.43 |
| | | 0% | 18% | 0% | 4% | 0% | 0% | 0% | 3% |
| | Type 3 | 0 | 0 | 0 | 8 | 0 | 0 | 0 | 1.14 |
| | | 0% | 0% | 0% | 35% | 0% | 0% | 0% | 5% |
| | Type 4 | 0 | 2 | 1 | 7 | 0 | 2 | 0 | 1.71 |
| | | 0% | 18% | 9% | 30% | 0% | 11% | 0% | 10% |
| **RE profiles** | Type 1 | 1 | 13 | 10 | 13 | 14 | 14 | 5 | 10 |
| | | 50% | 65% | 100% | 45% | 93% | 88% | 100% | 77% |
| | Type 2 | 0 | 1 | 0 | 2 | 0 | 0 | 0 | 0.43 |
| | | 0% | 5% | 0% | 7% | 0% | 0% | 0% | 2% |
| | Type 3 | 0 | 3 | 0 | 11 | 1 | 1 | 0 | 2.29 |
| | | 0% | 15% | 0% | 38% | 7% | 6% | 0% | 9% |
| | Type 4 | 1 | 3 | 0 | 3 | 0 | 1 | 0 | 1.14 |
| | | 50% | 15% | 0% | 10% | 0% | 6% | 0% | 12% |

task description shown to participants before they start their evaluation. Additionally, we introduced a competence classification to distinguish general competences from technical and research competences. However, to accommodate this classification we dropped the previous assessment of competence levels, as we did not want to double the workload of our study participants.

### *Automatic generation of online surveys*

For our revised experiment, we set up a web-based user profile evaluation system. In the new set up, instead of generating LaTeX profiles for users, we implemented a survey-style profile generation tool that queries the populated knowledge base and generates web-based profiles compatible with LimeSurvey (https://www.limesurvey.org), an open source survey application with built-in analytics features, as shown in Fig. 6. Similar to the first experiment, we generated two surveys for each user: one with the competence topics extracted from the full-text of documents and one with topics extracted from the rhetorical zones only. To lessen the *priming bias*—where participants may think topics shown earlier in the survey must be more relevant to them—we randomized the order of survey questions and informed the users in the evaluation instructions about this fact. However, we internally kept the original rank of the competence topics shown in survey questions as they appear in the knowledge base profiles, so that we can compute the precision of our system in top-$k$ cut-off ranks. We invited 32 computer scientists to participate in our user evaluations. In total, 25 users responded to the survey (note that an anonymized user like 'R1' from the second study is not necessarily the same person as in the first study). In contrast to the

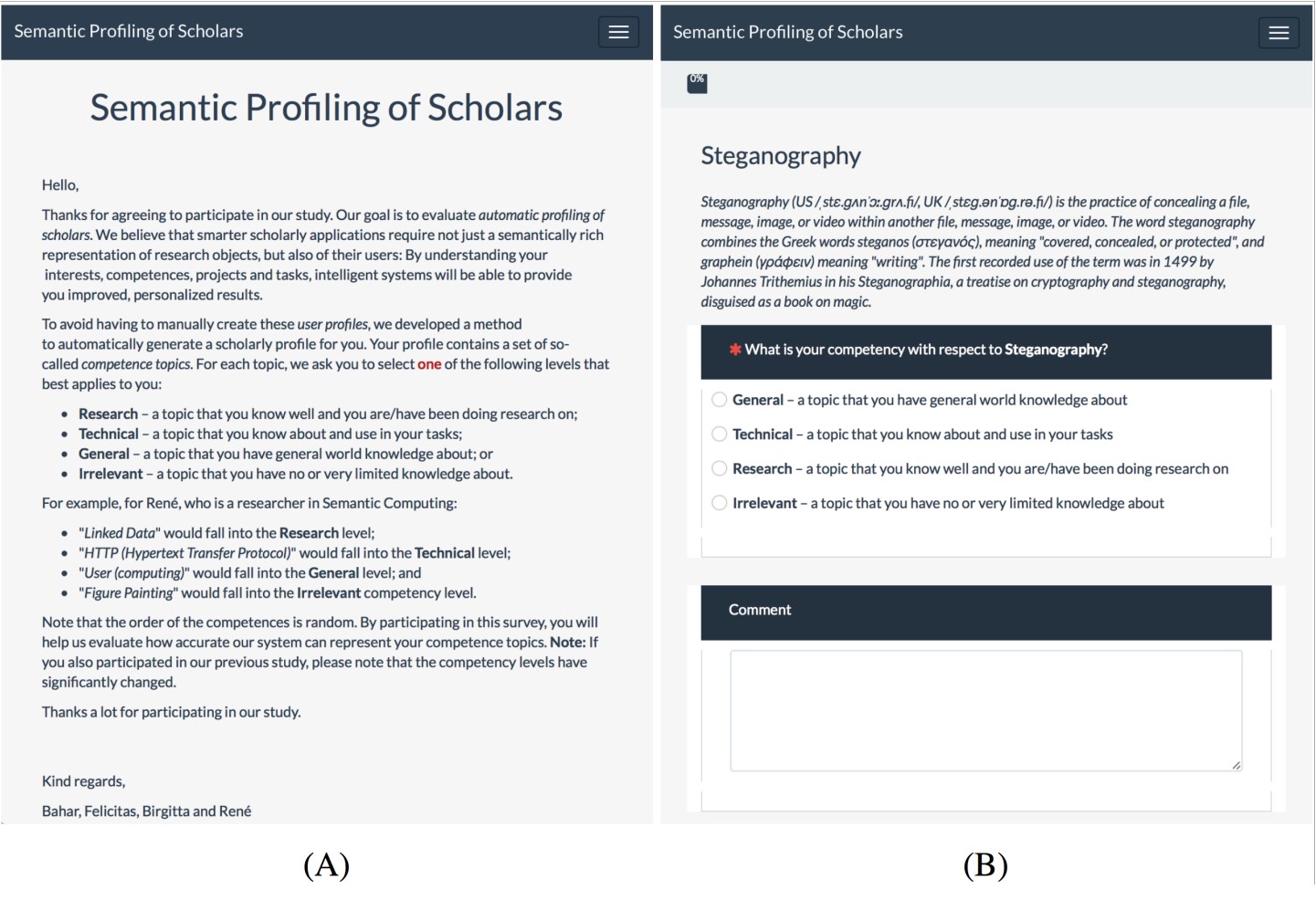

**Figure 6** An automatically generated web-based survey using *LimeSurvey*: (A) depicts the instructions shown to participants to explain the survey motivation and how to select a competence level, and (B) shows an example competency question with an interactive response interface.

previous survey, this time we asked the users to rate the competences along three different competence types, namely *General* which comprises very general and broad topics such as "*System*" or "*Number*", *Technical* which refers to skills a computer scientist needs in daily work, e.g., "*Hypertext Transfer Protocol*", and *Research*, which points to research topics a user has been or currently is involved in, e.g., "*Linked Data*".

### Result computation

All responses were exported into comma-separated value (CSV) files and analyzed with our own Java-based command-line tool, transforming the original horizontal schema into a vertical structure, based on their original rank. We computed the Precision@Rank, the Mean Average Precision (MAP) and the normalized Discounted Cumulative Gain (nDCG), according to the equations presented above ('Methodology and Metrics'). Table 5 presents the responses for both versions, the full-text profiles and RE Zones, with respect to the overall ratings across the four different competence levels. The results for the precision metrics are displayed in Tables 6 and 7.

**Table 5  Analysis of the survey responses for profiles generated from Full-text and RE Zones.** The values shown in the columns are number of competence types as voted by the survey participants.

| User | Competence type in full-text | | | | Competence type in RE zones | | | |
|------|---------|-----------|----------|------------|---------|-----------|----------|------------|
|      | General | Technical | Research | Irrelevant | General | Technical | Research | Irrelevant |
| R1   | 19 | 7  | 13 | 11 | 17 | 10 | 8  | 15 |
| R2   | 6  | 21 | 18 | 5  | 9  | 21 | 15 | 5  |
| R3   | 14 | 16 | 12 | 8  | 9  | 9  | 9  | 23 |
| R6   | 24 | 8  | 15 | 3  | 12 | 10 | 13 | 15 |
| R9   | 13 | 18 | 9  | 10 | 7  | 25 | 6  | 12 |
| R10  | 12 | 23 | 13 | 2  | 18 | 18 | 5  | 9  |
| R11  | 17 | 19 | 13 | 1  | 20 | 17 | 9  | 4  |
| R12  | 12 | 16 | 21 | 1  | 19 | 13 | 16 | 2  |
| R14  | 11 | 23 | 10 | 6  | 16 | 19 | 6  | 9  |
| R15  | 12 | 23 | 8  | 7  | 16 | 18 | 3  | 13 |
| R16  | 14 | 18 | 15 | 3  | 15 | 22 | 10 | 3  |
| R17  | 16 | 16 | 12 | 6  | 20 | 18 | 8  | 4  |
| R18  | 5  | 12 | 30 | 3  | 15 | 22 | 13 | 0  |
| R19  | 8  | 20 | 15 | 7  | 9  | 14 | 18 | 9  |
| R21  | 3  | 9  | 36 | 2  | 4  | 13 | 32 | 1  |
| R23  | 22 | 18 | 8  | 2  | 18 | 18 | 9  | 5  |
| R25  | 10 | 23 | 10 | 7  | 14 | 15 | 12 | 9  |
| R26  | 18 | 15 | 8  | 9  | 22 | 9  | 6  | 13 |
| R27  | 16 | 14 | 13 | 7  | 12 | 19 | 13 | 6  |
| R28  | 2  | 27 | 18 | 3  | 4  | 24 | 18 | 4  |
| R29  | 6  | 8  | 22 | 14 | 6  | 15 | 12 | 17 |
| R30  | 13 | 21 | 12 | 4  | 22 | 6  | 7  | 15 |
| R31  | 9  | 19 | 14 | 8  | 14 | 14 | 11 | 11 |
| R35  | 7  | 7  | 31 | 5  | 5  | 19 | 18 | 8  |
| R36  | 17 | 9  | 17 | 7  | 9  | 9  | 17 | 15 |
|      | | | | Total | | | | |
|      | 306 | 410 | 393 | 141 | 332 | 397 | 294 | 227 |
|      | 24.48% | 32.80% | 31.44% | 11.28% | 26.56% | 31.76% | 23.52% | 18.16% |

Since Precision@k and MAP are based on binary ratings (relevant/non-relevant), it has to be specified which competence levels to take into account. Therefore, we defined two thresholds: *Irrelevant* (Threshold '0') and *General* (Threshold '1'). For threshold '0', we treated the responses in the *General*, *Technical* and *Research* competence types as relevant. In this case, only *Irrelevant* entries are counted as errors. However, this might not be appropriate for every application: some use cases might want to also exclude competences in the *General* category. Therefore, we also analyzed the results for ratings above *General*, in order to ensure an equal distribution (Tables 6 and 7). Here, competences were only considered as relevant when they were rated either as *Technical* or *Research*.

Additionally, we computed the nDCG for each profile, which does not penalize for irrelevant competence topics in profiles.

**Table 6  Precision computation for profiles generated from full-text with a relevance threshold of *Irrelevant (0)* and *General (1)*.** All ratings above *Irrelevant (0)* and *General (1)* have been considered as relevant, respectively.

| User | Threshold 0—Irrelevant | | | | | | Threshold 1—General | | | | | | nDCG |
|------|------|------|------|------|------|------|------|------|------|------|------|------|------|
| | Average precision | | | Precision@k | | | Average precision | | | Precision@k | | | |
| | AP@10 | AP@25 | AP@50 | P@10 | P@25 | P@50 | AP@10 | AP@25 | AP@50 | P@10 | P@25 | P@50 | |
| R1 | 0.90 | 0.87 | 0.85 | 0.80 | 0.88 | 0.78 | 0.88 | 0.69 | 0.63 | 0.60 | 0.52 | 0.40 | 0.90 |
| R2 | 0.87 | 0.86 | 0.88 | 0.80 | 0.88 | 0.90 | 0.75 | 0.79 | 0.80 | 0.70 | 0.84 | 0.78 | 0.88 |
| R3 | 1.00 | 0.90 | 0.88 | 1.00 | 0.84 | 0.84 | 0.88 | 0.80 | 0.75 | 0.90 | 0.68 | 0.56 | 0.92 |
| R6 | 1.00 | 0.99 | 0.96 | 1.00 | 0.96 | 0.94 | 0.37 | 0.45 | 0.45 | 0.40 | 0.44 | 0.46 | 0.83 |
| R9 | 0.95 | 0.89 | 0.85 | 0.90 | 0.80 | 0.80 | 0.70 | 0.63 | 0.58 | 0.60 | 0.52 | 0.54 | 0.80 |
| R10 | 1.00 | 1.00 | 0.99 | 1.00 | 1.00 | 0.96 | 0.96 | 0.89 | 0.84 | 0.9 | 0.8 | 0.72 | 0.93 |
| R11 | 1.00 | 1.00 | 1.00 | 1.00 | 1.00 | 0.98 | 0.90 | 0.81 | 0.76 | 0.80 | 0.72 | 0.64 | 0.92 |
| R12 | 1.00 | 1.00 | 1.00 | 1.00 | 1.00 | 0.98 | 0.79 | 0.78 | 0.78 | 0.70 | 0.84 | 0.74 | 0.88 |
| R14 | 0.93 | 0.94 | 0.94 | 0.90 | 0.96 | 0.88 | 0.93 | 0.88 | 0.82 | 0.90 | 0.80 | 0.66 | 0.88 |
| R15 | 0.82 | 0.80 | 0.82 | 0.70 | 0.80 | 0.86 | 0.83 | 0.70 | 0.64 | 0.60 | 0.60 | 0.62 | 0.87 |
| R16 | 1.00 | 1.00 | 0.98 | 1.00 | 1.00 | 0.94 | 1.00 | 0.90 | 0.83 | 0.9 | 0.8 | 0.66 | 0.95 |
| R17 | 1.00 | 0.94 | 0.91 | 0.90 | 0.92 | 0.88 | 0.83 | 0.73 | 0.66 | 0.60 | 0.64 | 0.56 | 0.89 |
| R18 | 1.00 | 0.98 | 0.97 | 1.00 | 0.96 | 0.94 | 0.79 | 0.86 | 0.86 | 0.90 | 0.88 | 0.84 | 0.94 |
| R19 | 1.00 | 0.96 | 0.91 | 1.00 | 0.88 | 0.86 | 0.82 | 0.72 | 0.70 | 0.70 | 0.64 | 0.70 | 0.88 |
| R21 | 1.00 | 0.96 | 0.95 | 0.90 | 0.96 | 0.96 | 1.00 | 0.96 | 0.94 | 0.90 | 0.96 | 0.90 | 0.97 |
| R23 | 0.99 | 0.96 | 0.96 | 0.90 | 0.96 | 0.96 | 0.61 | 0.62 | 0.60 | 0.70 | 0.56 | 0.52 | 0.84 |
| R25 | 0.93 | 0.86 | 0.85 | 0.90 | 0.84 | 0.86 | 0.92 | 0.81 | 0.75 | 0.80 | 0.72 | 0.66 | 0.89 |
| R26 | 0.93 | 0.92 | 0.90 | 0.90 | 0.88 | 0.82 | 0.67 | 0.58 | 0.55 | 0.50 | 0.52 | 0.46 | 0.84 |
| R27 | 0.81 | 0.83 | 0.86 | 0.80 | 0.84 | 0.86 | 0.77 | 0.68 | 0.63 | 0.70 | 0.52 | 0.54 | 0.86 |
| R28 | 1.00 | 0.97 | 0.94 | 1.00 | 0.88 | 0.94 | 1.00 | 0.97 | 0.94 | 1.00 | 1.00 | 0.88 | 0.97 |
| R29 | 0.92 | 0.83 | 0.79 | 0.80 | 0.72 | 0.72 | 0.75 | 0.70 | 0.67 | 0.70 | 0.6 | 0.6 | 0.86 |
| R30 | 0.91 | 0.89 | 0.91 | 0.90 | 0.92 | 0.92 | 0.54 | 0.60 | 0.64 | 0.50 | 0.68 | 0.66 | 0.85 |
| R31 | 0.95 | 0.93 | 0.89 | 0.90 | 0.92 | 0.84 | 0.71 | 0.72 | 0.71 | 0.70 | 0.76 | 0.66 | 0.87 |
| R35 | 0.79 | 0.88 | 0.90 | 0.90 | 0.88 | 0.86 | 0.77 | 0.80 | 0.79 | 0.80 | 0.76 | 0.76 | 0.90 |
| R36 | 0.99 | 0.91 | 0.88 | 0.90 | 0.88 | 0.86 | 0.99 | 0.91 | 0.88 | 0.90 | 0.88 | 0.86 | 0.94 |
| | Mean average precision | | | Average | | | Mean average precision | | | Average | | | |
| | 0.95 | 0.92 | 0.91 | 0.91 | 0.91 | 0.89 | 0.80 | 0.76 | 0.73 | 0.74 | 0.70 | 0.66 | 0.89 |

### Discussion

Overall, compared to our first user study, our enhanced method resulted in fewer irrelevant results in the user profiles. This is partially due to the improvements mentioned above, where we removed irrelevant entries that affected every generated profile (e.g., the competency entries derived from the word "*figure*").

We also analyzed the distribution of competence topic types in each user profile (Fig. 7). In both profile versions, about 55–65% of the detected competences were rated either as *Technical* or *Research*, which corroborates our hypothesis that the named entities in users' publications are representative of their research expertise. In comparison with the full-text and RE-only version of each user profile, although we observe an increase in the number of irrelevant topics, majority of them fall into the *Research* and *Technical* types.

**Table 7 Precision computation for profiles generated from RE zones with a relevance threshold of *Irrelevant (0)* and *General (1)*.** All ratings above *Irrelevant (0)* and *General (1)* have been considered as relevant, respectively.

| User | Threshold 0—irrelevant | | | | | | Threshold 1—general | | | | | | nDCG |
|------|---------|---------|---------|---------|---------|---------|---------|---------|---------|---------|---------|---------|------|
| | Average precision | | | Precision | | | Average precision | | | Precision | | | |
| | AP@10 | AP@25 | AP@50 | P@10 | P@25 | P@50 | AP@10 | AP@25 | AP@50 | P@10 | P@25 | P@50 | |
| R1 | 0.93 | 0.90 | 0.86 | 0.90 | 0.80 | 0.70 | 0.84 | 0.68 | 0.56 | 0.50 | 0.40 | 0.36 | 0.85 |
| R2 | 0.87 | 0.83 | 0.86 | 0.70 | 0.88 | 0.90 | 0.86 | 0.83 | 0.79 | 0.70 | 0.80 | 0.72 | 0.88 |
| R3 | 0.99 | 0.93 | 0.87 | 0.90 | 0.80 | 0.54 | 0.98 | 0.87 | 0.81 | 0.80 | 0.60 | 0.36 | 0.90 |
| R6 | 0.98 | 0.91 | 0.83 | 0.90 | 0.76 | 0.70 | 0.60 | 0.60 | 0.55 | 0.60 | 0.52 | 0.46 | 0.83 |
| R9 | 0.91 | 0.80 | 0.79 | 0.70 | 0.80 | 0.76 | 0.79 | 0.66 | 0.64 | 0.60 | 0.64 | 0.62 | 0.81 |
| R10 | 0.99 | 0.94 | 0.90 | 0.90 | 0.88 | 0.82 | 0.83 | 0.75 | 0.62 | 0.70 | 0.48 | 0.46 | 0.87 |
| R11 | 1.00 | 1.00 | 0.95 | 1.00 | 0.96 | 0.92 | 0.99 | 0.86 | 0.71 | 0.80 | 0.56 | 0.52 | 0.90 |
| R12 | 1.00 | 1.00 | 0.99 | 1.00 | 1.00 | 0.96 | 0.67 | 0.60 | 0.60 | 0.50 | 0.56 | 0.58 | 0.88 |
| R14 | 1.00 | 0.97 | 0.90 | 1.00 | 0.84 | 0.82 | 0.72 | 0.71 | 0.64 | 0.80 | 0.60 | 0.5 | 0.87 |
| R15 | 0.99 | 0.90 | 0.84 | 0.90 | 0.80 | 0.74 | 0.70 | 0.65 | 0.59 | 0.70 | 0.56 | 0.42 | 0.82 |
| R16 | 1.00 | 0.99 | 0.96 | 1.00 | 0.92 | 0.94 | 0.95 | 0.88 | 0.76 | 0.90 | 0.64 | 0.64 | 0.91 |
| R17 | 1.00 | 1.00 | 0.98 | 1.00 | 0.96 | 0.92 | 0.88 | 0.81 | 0.71 | 0.80 | 0.64 | 0.52 | 0.90 |
| R18 | 1.00 | 1.00 | 1.00 | 1.00 | 1.00 | 1.00 | 0.86 | 0.80 | 0.73 | 0.80 | 0.68 | 0.70 | 0.93 |
| R19 | 1.00 | 0.99 | 0.93 | 1.00 | 0.88 | 0.82 | 0.75 | 0.70 | 0.69 | 0.50 | 0.68 | 0.64 | 0.89 |
| R21 | 1.00 | 1.00 | 1.00 | 1.00 | 1.00 | 0.98 | 1.00 | 1.00 | 0.99 | 1.00 | 1.00 | 0.9 | 0.95 |
| R23 | 1.00 | 0.97 | 0.95 | 1.00 | 0.92 | 0.90 | 0.78 | 0.76 | 0.71 | 0.80 | 0.68 | 0.54 | 0.88 |
| R25 | 1.00 | 0.88 | 0.85 | 0.80 | 0.84 | 0.82 | 0.98 | 0.77 | 0.69 | 0.70 | 0.60 | 0.54 | 0.93 |
| R26 | 0.98 | 0.88 | 0.80 | 0.90 | 0.72 | 0.74 | 0.61 | 0.53 | 0.41 | 0.40 | 0.28 | 0.30 | 0.80 |
| R27 | 1.00 | 0.97 | 0.93 | 1.00 | 0.88 | 0.88 | 0.95 | 0.89 | 0.79 | 0.90 | 0.68 | 0.64 | 0.93 |
| R28 | 0.84 | 0.84 | 0.87 | 0.90 | 0.84 | 0.92 | 0.84 | 0.83 | 0.82 | 0.90 | 0.76 | 0.84 | 0.90 |
| R29 | 0.60 | 0.66 | 0.67 | 0.60 | 0.72 | 0.66 | 0.53 | 0.58 | 0.56 | 0.50 | 0.64 | 0.54 | 0.76 |
| R30 | 0.82 | 0.81 | 0.77 | 0.70 | 0.76 | 0.70 | 0.83 | 0.53 | 0.45 | 0.30 | 0.36 | 0.26 | 0.84 |
| R31 | 0.96 | 0.86 | 0.81 | 0.80 | 0.72 | 0.78 | 0.64 | 0.57 | 0.51 | 0.50 | 0.48 | 0.50 | 0.81 |
| R35 | 1.00 | 0.88 | 0.84 | 1.00 | 0.88 | 0.84 | 1.00 | 0.92 | 0.83 | 1.00 | 0.76 | 0.74 | 0.92 |
| R36 | 1.00 | 0.94 | 0.85 | 0.90 | 0.84 | 0.70 | 0.95 | 0.86 | 0.77 | 0.90 | 0.88 | 0.86 | 0.94 |
| | Mean average precision | | | Average | | | Mean average precision | | | Average | | | |
| | 0.95 | 0.91 | 0.88 | 0.90 | 0.86 | 0.82 | 0.82 | 0.74 | 0.68 | 0.70 | 0.62 | 0.57 | 0.88 |

These results are also consistent with our hypothesis that the topics in rhetorical zones of scholarly literature, like claims and contributions of authors are strong indications of their competence. As we can see from the results, the full-text profiles returned less irrelevant results and more higher ratings (64%) than the RE-only version (55%). A closer look on individual responses revealed that the error (Type 1 (Wrong URI)) occurred more often in the RE-only version. A wrong matching of extracted terms to URIs mainly causes a wrong description and hence an irrelevant result. Longer and more comprehensive text passages, as in the full-text profiles, might better compensate this problem and therefore result in less URI mismatches. Too broad and general competences are a further issue when looking at the ratings. Again, the reason was that DBpedia Spotlight that does not distinguish

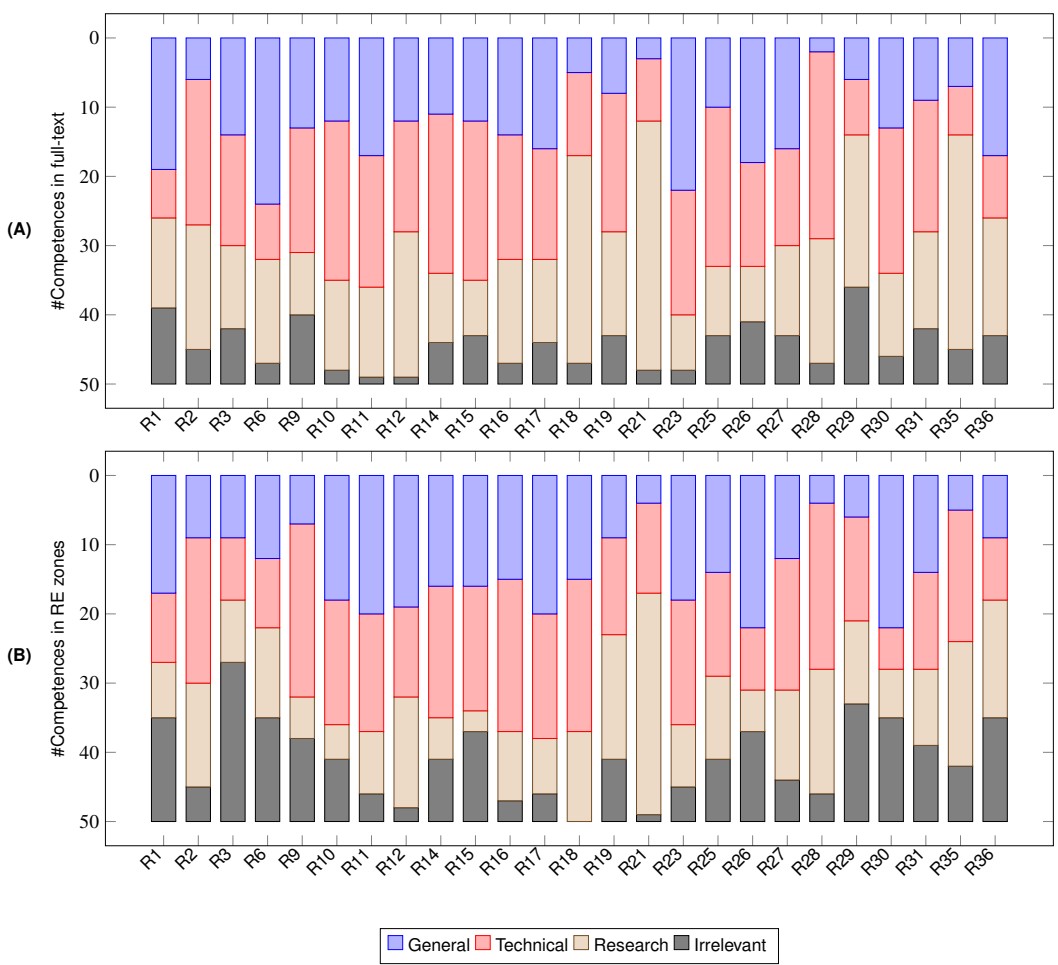

**Figure 7** The two plots show the distribution of top-50 competence types in full-text (A) and RE-only (B) profiles from the evaluation survey responses.

between composite and single terms, for instance, "*service*" and "*service provider*". It finds successful matches for both terms and thus produces general topics.

We also evaluated the rated competences with respect to their ranking in the result list. Both metrics, Precision@k and Mean Average Precision have been computed across two relevance thresholds. *Threshold '0'* denotes the results where all ratings above *Irrelevant* were considered as relevant, namely, *General, Technical and Research*. Since this division favours relevant competences, we additionally computed the Precision@k and Mean Average Precision for ratings above *General*. Among the top-10 results, the RE-only profiles performed slightly better for *Threshold '1'*, which indicates that all the relevant competences are a bit more likely in the Top-10 results than in the full-text profiles. However, the ranking results turn upside down for the Top-50 results, where the MAP value for the full-text version is significantly higher than for the RE-only version. That reveals the ranking in full-text profiles is more stable over 50 competences compared to RE zones.

Additionally, we analyzed whether the different number of papers per user has an influence on the results. It turned out that there is no correlation between the number of papers used and the obtained precision. Furthermore, we neither take into account a decay function nor distinguish between recent publications or papers a user has written a longer time ago. This leaves room for future work. For all our results, it also needs to be considered that we did not ask the users about missing competences and therefore we did not penalize for incomplete results. The results are grounded on relevance assessments for automatically extracted competences from publications. Rather than completeness, it is an approach to counteract the cold-start problem in personalized applications and to minimize the burden for the user to explicitly provide pertinent information.

Overall, we reached high ranking values in the top-10 results for both thresholds, the MAP values for *Research* and *Technical* competences vary between 0.80 and 0.95. Looking at the Top-50 competences, full-text profiles performed better than RE zones. Hence, we can conclude that our approach is effective for detecting a user's background knowledge implicitly.

### Threats to validity

A threat to the external validity of our findings is the scale of evaluation, generalizing the results obtained from studying a relatively small group of computer scientists. To mitigate this effect, when choosing our sample group, we tried to be inclusive of researchers from various computer science sub-disciplines, including those who work in inter-disciplinary areas, such as bioinformatics. We also tried to include both young researchers and senior scientists with a larger number of selected publications. Although the evaluation is admittedly not definitive, our results show the effectiveness and accuracy of our method for the automatic creation of scholarly profiles. At its current stage, we believe that the open source implementation of our approach will facilitate others to collect further empirical evidence, by integrating ScholarLens into various scholarly applications.

## APPLICATION EXAMPLES

As one of our main contributions in this paper is an open source library for generating semantic user profiles from scholarly articles, we want to demonstrate how various end-user applications can incorporate the generated user profiles. Towards this end, in this section we present a number of use cases for exploiting the knowledge base generated by *ScholarLens*.

### Finding all competences of a user

By querying the populated knowledge base with the researchers' profiles, we can find all topics that a user is competent in. Following our knowledge base schema (see 'Design'), we can query all the competence records of a given author URI and find the topics (in form of LOD URIs), from either the papers' full-text or exclusively the RE zones. In fact, the SPARQL query shown below is how we gathered each user's competences (from RE zones) to generate the evaluation profiles described in 'Evaluation':

**Table 8  Example response for finding all competence topics of a user. The 'Extracted Competence Topics' column lists the topics as linked entities on the web of LOD.** The competence topics are sorted by their raw frequency in the user profile, in descending order.

| User | Extracted competence topics |
| --- | --- |
| R1 | dbpedia:Tree_(data_structure), dbpedia:Vertex_(graph_theory), dbpedia:Cluster_analysis, … |
| R2 | dbpedia:Natural_language_processing, dbpedia:Semantic_Web, dbpedia:Entity-relationship_model, … |
| R3 | dbpedia:Recommender_system, dbpedia:Semantic_web, dbpedia:Web_portal, dbpedia:Biodiversity, … |
| R4 | dbpedia:Service_(economics), dbpedia:Feedback, dbpedia:User_(computing), dbpedia:System, … |
| R5 | dbpedia:Result, dbpedia:Service_discovery, dbpedia:Web_search_engine, dbpedia:Internet_protocol, … |

```
SELECT DISTINCT ?uri (COUNT(?uri) AS ?count) WHERE {
    ?creator rdf:type um:User .
    ?creator rdfs:isDefinedBy <http://semanticsoftware.info/lodexporter/creator/R1> .
    ?creator um:hasCompetencyRecord ?competenceRecord .
    ?competenceRecord c:competenceFor ?competence .
    ?competence rdfs:isDefinedBy ?uri .
    ?rhetoricalEntity rdf:type sro:RhetoricalElement .
    ?rhetoricalEntity pubo:containsNE ?competence .
} GROUP BY ?uri ORDER BY DESC(?count)
```

Table 8 shows a number of competence topics (grounded to their LOD URIs) for some of our evaluation participants, sorted in descending order by their frequency in the documents.

### Ranking papers based on a user's competences

Semantic user profiles can be incredibly effective in the context of information retrieval systems. Here, we demonstrate how they can help to improve the relevance of the results. Our proposition is that papers that mention the competence topics of a user are more *interesting* for her and thus, should be ranked higher in the results. Therefore, the diversity and frequency of topics within a paper should be used as ranking features. We showed in *Sateli & Witte (2015)* that retrieving papers based on their LOD entities is more effective than conventional keyword-based methods. However, the results were not presented in order of their *interestingness* for the end-user. Here, we integrate our semantic user profiles to re-rank the results, based on the common topics in both the papers and a user's profile:

```
SELECT (COUNT(DISTINCT ?uri) as ?rank) WHERE {
    <http: // example.com/example_paper.xml> pubo:hasAnnotation ?topic .
    ?topic   rdf:type   pubo:LinkedNamedEntity .
    ?topic   rdfs:isDefinedBy  ? uri  .
FILTER EXISTS {
    ? creator   rdfs:isDefinedBy   <http: // semanticsoftware . info / lodexporter
        / creator /R8> .
    ? creator   um:hasCompetencyRecord ?competenceRecord .
    ?competenceRecord c:competenceFor ?competence .
    ?competence  rdfs:isDefinedBy  ? uri  .}
}
```

The query shown above compares the topic URIs in a given paper to user R8's competences extracted from full-text documents and counts the occurrence of such a hit. Note that the `DISTINCT` keyword will cause the query to only count the unique topics, e.g., if `<dbpedia:Semantic_Web>` appears two times in the paper, it will be counted as one occurrence. We decided to count the unique occurrences, because a ranking algorithm based on the raw frequency of competence topics will favour long (non-normalized) papers over shorter ones. We can then use the numbers returned by the query above as a means to rank the papers. Table 9 shows the result set returned by performing a query against the SePublica dataset of 29 papers from (*Sateli & Witte, 2015*) to find papers mentioning `<dbpedia:Ontology_(information_science)>`. The "*Topic Mentions*" column shows the ranked results based on how many times the query topic was mentioned in a document. In contrast, the R6 and R8 profile-based columns show the ranked results using the number of common topics between the papers (full-text) and the researchers' respective profiles (populated from their own publications full-text). Note that in the R6 and R8 profile-based columns, we only count the number of unique topics and not their frequency. An interesting observation here is that the paper ranked fourth in the frequency-based column ranks last in both profile-based result sets. A manual inspection of the paper revealed that this document, although originally ranked high in the results, is in fact an editors' note in the preface of the SePublica 2012 proceedings. On the other hand, the paper which ranked first in the frequency-based column, remained first in R8's result set, since this user has a stronger research focus on ontologies and linked open data compared to R6, as we observed from their generated profiles during evaluation.

### Finding users with related competences

Given the semantic user profiles and a topic in form of an LOD URI, we can find all users in the knowledge base that have related competences. By virtue of traversing the LOD cloud, we can find topic URIs that are (semantically) related to a given competence topic and match against users' profiles to find competent authors:

**Table 9** Personalized re-ranking of search results: this tables shown how integration of semantic user profiles for researchers R6 and R8 affects ranking of the top-10 results originally retrieved and sorted by a frequency-based method.

| Paper title | Topic mentions | | R8's Profile | | R6's Profile | |
|---|---|---|---|---|---|---|
| | Rank | Raw frequency | Rank | Com. topics | Rank | Com. topics |
| "*A Review of Ontologies for Describing Scholarly and Scientific Documents*" | 1 | 92 | 1 | 312 | 5 | 198 |
| "*BauDenkMalNetz—Creating a Semantically Annotated Web Resource of Historical Buildings*" | 2 | 50 | 5 | 294 | 4 | 203 |
| "*Describing bibliographic references*" in RDF | 3 | 38 | 6 | 269 | 8 | 177 |
| "*Semantic Publishing of Knowledge about Amino Acids*" | 4 | 25 | 10 | 79 | 10 | 53 |
| "*Supporting Information Sharing for Re-Use and Analysis of Scientific Research Publication Data*" | 5 | 25 | 4 | 306 | 7 | 185 |
| "*Linked Data for the Natural Sciences: two Use Cases in Chemistry and Biology*" | 6 | 23 | 2 | 310 | 1 | 220 |
| "*Ornithology Based on Linking Bird Observations with Weather Data*" | 7 | 22 | 8 | 248 | 6 | 189 |
| "*Systematic Reviews as an Interface to the Web of (Trial) Data: using PICO as an Ontology for Knowledge Synthesis in Evidence-based Healthcare Research*" | 8 | 19 | 9 | 179 | 9 | 140 |
| "*Towards the Automatic Identification of the Nature of Citations*" | 9 | 19 | 3 | 307 | 2 | 214 |
| "*SMART Research using Linked Data—Sharing Research Data for Integrated Water Resources Management in the Lower Jordan Valley*" | 10 | 19 | 7 | 260 | 3 | 214 |

```
PREFIX dcterms: <http: // purl .org/dc/terms/>
PREFIX dbpedia: <http: // dbpedia.org/ resource />

SELECT ?author_uri WHERE {
    SERVICE <http://dbpedia.org/ sparql > {
        dbpedia:Ontology_( information_science )  dcterms:subject  ? category  .
        ? subject    dcterms:subject  ? category  .
    }
    ?author  rdf:type  um:User .
    ? creator  rdfs:isDefinedBy  ? author_uri  .
    ? creator  um:hasCompetencyRecord ?competenceRecord.
    ?competenceRecord c:competenceFor ?competence.
    ?competence rdfs:isDefinedBy  ? subject .
    ? rhetoricalEntity  pubo:containsNE ?competence.
    ? rhetoricalEntity   rdf:type   sro:RhetoricalElement .
}
```

The query above first performs a federated query against DBpedia's SPARQL endpoint to find topic URIs that are semantically related to the query topic (we assume all topics under the same *category* in the DBpedia ontology are semantically related). Then, it matches the retrieved URIs against the topics of the knowledge base users'

**Table 10** By virtue of traversing the web of LOD, topics semantically-related to the query are inferred and their respective competent researchers are shown in the 'Competent Users' column.

| Competence topic | Competent users |
|---|---|
| dbpedia:Ontology_(information_science) | R1, R2, R3, R8 |
| dbpedia:Linked_data | R2, R3, R8 |
| dbpedia:Knowledge_representation_and_reasoning | R1, R2, R4, R8 |
| dbpedia:Semantic_Web | R1, R2, R3, R4, R5, R6, R7, R8 |
| dbpedia:Controller_vocabulary | R2, R3, R8 |
| dbpedia:Tree_(data_structure) | R1, R4, R7 |

competence records. As shown in Table 10, with this approach we can find a researcher competent in ⟨dbpedia:Ontology_(information_science)⟩, even when this topic is not mentioned directly in her profile, but only has ⟨dbpedia:Linked_data⟩, since both of the aforementioned topics are related in the DBpedia ontology. In other words, if we are looking for persons competent in ontologies, a researcher that has previously conducted research on linked data might also be a suitable match.

## CONCLUSIONS

We presented semantic user profiles as the next important extension for semantic publishing applications: with a standardized, shareable, and extendable representation of a user's competences, a number of novel application scenarios now becomes possible. For example, searching for scientists with specific competences can help to find reviewers for a given paper or proposal. Recommendation algorithms can filter and rank the immense amount of research objects, based on the profile of an individual user. A wealth of additional applications becomes feasible, such as matching the competences of a research group against project requirements, simply by virtue of analyzing an inter-linked knowledge graph of users, datasets, publications, and other artifacts.

We showed how these ideas can be realized with semantic scholarly profiles that are based on a Linked Open Data-compliant format. To enable the next generation of scholarly applications, we developed a method for the automatic construction of open knowledge bases with semantic user profiles. Our method is implemented in form of the open source *ScholarLens* library, which can be easily integrated into existing or new scholarly systems. An important part of our work is the automatic generation of the user profiles through a text mining pipeline, which helps to overcome the well-known cold start problem in user profiling. Unlike other existing approaches, ScholarLens populates a knowledge base compliant with the web of linked data, which facilitates connecting the semantic user profiles with other domain- and application-specific information. We demonstrated how such a knowledge base can be exploited for various use cases, using standard SPARQL queries.

Based on the results of two rounds of user studies, we can conclude that the generated profiles represent the competences of scientists with a very high accuracy. Evaluating the impact of these profiles on different, concrete tasks is the next logical step in this research.

Towards this end, we are currently integrating our profile knowledge base into a scholarly data portal for biodiversity research, in order to evaluate their impact on concrete research questions in a life sciences scenario.

### Funding

This work was partially funded by an NSERC Discovery Grant. This work was also supported by the DAAD (German Academic Exchange Service) through the PPP Canada program and by the DFG (German Research Foundation) within the GFBio project. There was no additional external funding received for this study. The funders had no role in study design, data collection and analysis, decision to publish, or preparation of the manuscript.

### Competing Interests

The authors declare there are no competing interests.

### Author Contributions

- Bahar Sateli, Felicitas Löffler and René Witte conceived and designed the experiments, performed the experiments, analyzed the data, contributed reagents/materials/analysis tools, wrote the paper, prepared figures and/or tables, performed the computation work, reviewed drafts of the paper.
- Birgitta König-Ries conceived and designed the experiments, wrote the paper, reviewed drafts of the paper.

### Data Availability

Semantic Software: http://www.semanticsoftware.info/semantic-user-profiling-peerj-2016-supplements.

GitHub: https://github.com/SemanticSoftwareLab/ScholarLens.

### Supplemental Information

Supplemental information for this article can be found online at http://dx.doi.org/10.7717/peerj-cs.121#supplemental-information.

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
