# Peer review of "ScholarLens: extracting competences from research publications for the automatic generation of semantic user profiles"

_PeerJ Computer Science, doi:10.7717/peerj-cs.121_

## Round 0.1 · original submission · Major Revisions

Please follow the reviewers' recommendations closely. Significant revisions are required before the paper ready for publication.

1. The problem addressed by the paper is interesting and important, but the contribution of the paper is not clear. As an application, the paper does not stand on its own, as the system is more of a proof-of-concept than a fieldable system. It is not clear whether or how the paper extends the state of the art in semantic technology or in methodology. A strong argument must be made as to why this work is worthy of publication and what communities will benefit from its publication.

2. The literature review is insufficient. In particular, please follow the reviewer suggestions as to comparing ScholarLens with other systems. How does it differ from other systems? What are the benefits relative to other systems? Please cite other NLP systems and provide a justification for the choice you made. Please address the reviewer's question on why you chose Apache Jena and whether alternatives might give better performance. Please provide better review of the literature on modeling experts.

3. The evaluation is clearly insufficient for a fielded system. Please justify why the small and not very diverse set of evaluators is adequate. Please address this question in the context of the paper's intended contribution. Please either provide clear justification for using yourselves in the evaluation or do an evaluation that does not include yourselves as users.

4. There needs to be a clear explanation, in the context of the literature review requested in item #2, of your methodology for modeling expertise.

Reviewer 1 ·

Basic reporting

This paper explains how to build semantic scholar profiles using ontologies and did user level evaluations.

Experimental design

The whole semantic scholar profile system is clearly described, but I did not see any innovation in advancing the state of the art in the domain of semantic web. It is a mere of applying existing technologies. It is also more like a proof-of-concept exercise, as the test data is relatively small.

For user evaluation, the sample group is too small, which also include authors themselves, which is highly bias. The online survey only has 20 researchers evaluate, which is problematic.

There is no comparison among different systems, such as compare your system with VIVO profiling,

Validity of the findings

The validity of the findings is questionable. To me, this is just a proof-of-concept exercise. It lacks of thorough large scale implementation and evaluation.The user evaluation at this small sample is fine, but definitely not ideal. To show the validity, it has to be large scale comparison with similar systems, and large scale user evaluation for at least hundreds or thousands.

Additional comments

In general, I think this paper is a proof-of-concept exercise.

It has the following issues:
- missing reference and evaluation with related systems: such as Aminer, Semantic Scholar, VIVO, Academia Search, Stanford profiling system, harvard also has its own. What are the special contribution/uniqueness from your research.
- how to model expertise, it is not clear in this article on how to address this issue.
- you always use GATE system, but there are many NLP tools, GATE is not the main stream, no justification is provided why GATE has been chosen for NLP tasks
- if you store your triples, Apache Jena is slow, did you try others, how many triples in your system. Is your system just a mock-up system?

·

Basic reporting

The article is well written and covers and interesting and important topic that has potentially important and significant impact. The introduction provides a good background to the area and the previous literature and work by the authors. The template used is effective and the figures are useful, though I think they could be made more 'stand alone' to encourage more readership. Relevant data is provided, subject to anonymity concerns, The work represents and improvement of work previously published by the authors as a conference report, but I think the additional work is significant.

Experimental design

The overall design and executions of the work is sensible and addresses the issues of automated completion of semantic profiles, is described in detail, clearly and effectively and ethical standards look to have been maintained.

Validity of the findings

The data reported is interesting and useful and reasonable conclusions reported and has relatively limited numbers of people involved. My concern arrises only from the fact that the group surveyed is purely from one discipline. I understand why this has been done but I am concerned that the real use of the approaches taken by the authors apply when the the system would be used at scale across a wide range of disciplines as the semantic profiles will be most effective when used to find people in different disciplines to the searcher but having complimentary skills needed for an example for an interdisciplinary project. I think it would be useful of the authors could add thoughts and any information they may have in this respect.

Additional comments

No comments

External reviews were received for this submission. These reviews were used by the Editor when they made their decision, and can be downloaded below.

---

## Round 0.2 · Minor Revisions

Your revisions addressed the reviewers' major concerns. The literature review and evaluation were significantly improved. A few minor issues remain. Please read the reviews carefully and address the issues presented. Specifically:

1. In your rebuttal letter, you stress that your main contribution is an open source library for creating semantic user profiles, and not a system per se. I understand this, but you do have more than just a library. You have both a method and a library for implementing the method. Not only do you downplay your contribution by limiting it to a "library," you find yourself writing ludicrous statements such as "The input to our library is a set of research articles." A system, process, pipeline or workflow has inputs. Each element of a library can have inputs. But a library itself does not have inputs or outputs, until its components have been arranged in some kind of workflow or process. Figure 1 describes "our approach" to semantic user profiling. Your contribution is the approach (which you need not argue is revolutionary if the main contribution is the open-source library) and an open-source library of components for accomplishing the approach. This framing is much stronger and avoids strange assertions about inputs to libraries.

2. Please clarify whether you are linking to or reusing the various LOD ontologies.

3. Please clarify the benefits of using SPAQL and RDF. Is this primarily to support open source solutions? For compatibility with your data sources?

4. Please comment briefly on your approach to modeling competence.

5. Reviewer 2 states: "the authors place extensive emphasis on the open source nature of the solution they provide and seem to imply that this is the main reason for publication., i.e. to make the code available even if the evidence that it will be really useful is not provided." However, Reviewer 1 still notes that "evaluation is superficial" and "it is hard to see whether it will be useful." I think the point should be strengthened by explicitly stating that the evaluation is admittedly not definitive, but it suggests that your approach has merit, and your aim is to make the library available for others to use and thereby collect further evidence in the future.

Reviewer 1 ·

Basic reporting

I think the authors made significant improvement this time, especially on literature review and evaluation.

Experimental design

The authors have made progress to improve the experimental design for evaluation.

Validity of the findings

Findings are still based on a small size of user evaluation

Additional comments

I think the authors made significant improvement this time, especially on literature review and evaluation.

Still some issues remained:
- how to model competence, this is a hard problem, there are many different ways of doing or attempting to model it formally. Domain taxonomies were used, such as MeSH, but still it is hard to model competence across difference domains, there is no integrated taxonomy across different domains. It is not clear how authors think about this issue. Other approaches are using keywords extracted from papers or their publications, which seems doable, again these applications are struggling with the accurate descriptions of competence.
- linked to LOD, it is not clear in this revised version, how they ontology are linked with other ontologies. I see that you are reusing many existing LOD ontologies, which are very good, but just link to FOAF, there is no much you can gain by connecting with LOD. Also most of the LOD datasets are not well maintained, except DBpedia. What are the benefits of linking to them?
- I understand your semantic approach of using RDF and SParql to query, but in this special case, what are the benefits of using RDF and Sparql (which sparql is very different to compose anyway). What is the ultimate goal in your application to use RDF and Sparql, it is not clear in your study.
- again, evaluation is good, but superficial. Using very small users, it is really hard to see whether your system will be useful or add value.

User profiling is a difficult problem. Google scholar, Microsoft academic search, and Aminer are doing such in different ways, but still struggle to provide better services. Your research is a good try, but it is more a proof-of-concept, it is far from a system which can work and add value.

·

Basic reporting

The paper is fine in the basic requirements.

Experimental design

This is within the scope of the journal. The evaluation that has been undertaken is methodologically sound but the concern is with the scope of the evaluation.

Validity of the findings

Within the relatively small scale and limited evaluation the findings are valid.

Additional comments

I was generally favourably disposed towards this paper but concerned with the limits evaluation. Despite these limits I would have agreed to publication even though proper evaluation would have resulted in a much stronger and more interesting paper. I have indicated my general suggestion for acceptance below) . However, in reply to the referees comments the authors place extensive emphasis on the open source nature of the solution they provide and seem to imply that this is the main reason for publication., i.e. to make the code available even if the evidence that it will be really useful is not provided.

Given this then I feel that it may be more appropriate simply to make the software available via GitHub rather than as a publication.

---

## Round 0.3 · accepted · Accept

The latest version of the paper has addressed the reviewers' concerns. I congratulate you on the paper's acceptance for publication.

I have one minor comment I hope you will address in your final version. On page 3, line 35, you say: “first open source library solving this task.” We can use the components of a library to solve a task, but the library is not the solution. Your method solves the task. Your library implements the method. Please reword this passage in your final edits.

Again, congratulations on your acceptance to PeerJ.